# PAE: Reinforcement Learning from External Knowledge for Efficient Exploration

**Zhe Wu**[* 1]**, Haofei Lu**[* 2]**, Junliang Xing**[† 2]**,**
**You Wu**[1,3]**, Renye Yan**[1,4]**, Yaozhong Gan** [1]**, Yuanchun Shi**[1,2]

[1]QiYuan Lab, [2]Department of Computer Science and Technology, Tsinghua University
[3]Nanjing University, [4]Peking University
{wuzhe, wuyou, yanrenye, ganyaozhong}@qiyuanlab.com,
luhf23@mails.tsinghua.edu.cn,{jlxing, shiyc}@tsinghua.edu.cn

## Abstract

Human intelligence is adept at absorbing valuable insights from external knowledge. This capability is equally crucial for artificial intelligence. In contrast, classical reinforcement learning agents lack such capabilities and often resort to extensive trial and error to explore the environment. This paper introduces **PAE**: **P**lanner-**A**ctor-**E**valuator, a novel framework for teaching agents to *learn to absorb external knowledge*. PAE integrates the Planner's knowledge-state alignment mechanism, the Actor's mutual information skill control, and the Evaluator's adaptive intrinsic exploration reward to achieve 1) effective cross-modal information fusion, 2) enhanced linkage between knowledge and state, and 3) hierarchical mastery of complex tasks. Comprehensive experiments across 11 challenging tasks from the BabyAI and MiniHack environment suites demonstrate PAE's superior exploration efficiency with good interpretability.

## 1 Introduction

Accepting suggestions from external guidance is an integral part of human learning. Humans can absorb insightful external knowledge and seamlessly integrate it into their strategies to tackle complex tasks (Council et al., 2000; Mills et al., 2010). By leveraging existing knowledge, human players can quickly adapt to completely unfamiliar games after just a few rounds of playing. Their ability to swiftly transfer and align existing knowledge with the current environment is a true testament to human intelligence. Building intelligent systems that can learn and integrate like humans is an ultimate goal we relentlessly pursue. Currently, reinforcement learning (RL) has demonstrated its potential towards this goal, with remarkable breakthroughs in numerous domains, approaching or even surpassing human capabilities on Atari Games (Mnih et al., 2013), Go (Silver et al., 2016) and StarCraft (Vinyals et al., 2019).

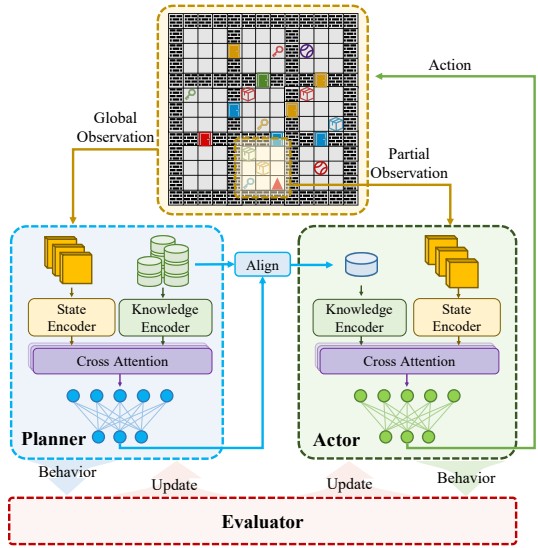

Figure 1: The Planner-Actor-Evaluator (PAE) framework consists of three key components.

However, currently RL struggles to efficiently capture and integrate the vast amount of expert knowledge already existing in the domain. Specifically, RL faces challenges in long-term and sparse reward tasks due to a lack of guidance, requiring extensive trial and error. Our primary focus in this

---

*Equal contribution
†Corresponding Author

work is teaching agents to leverage external knowledge and approach optimal solutions faster in sparse reward environments.

Although mimicking human capabilities benefits RL, it remains challenging for RL agents to realize the above capabilities (Chowdhury et al., 2023). Three main challenges arise when training agents to absorb external knowledge to improve their capabilities. **(1) Difficulties in knowledge acquisition and representation.** Existing approaches utilize Behavioral Cloning (BC) to acquire knowledge by imitating human behaviors directly (Codevilla et al., 2018) or adopt Inverse Reinforcement Learning (IRL) to learn reward functions for providing knowledge (Arora & Doshi, 2021). However, this necessitates many demonstrations, which are costly to label and collect. **(2) Obstacles to the integration of external knowledge and internal strategies.** Integration requires bridging the gap between the discrete nature of external knowledge and the continuous nature of internal strategies. Two potential directions emerge: one is enabling internal strategies to query external knowledge, but it faces challenges at the retrieval level (Reid et al., 2022). The other approach is directly encoding domain knowledge as propositional rules into neural networks for a warm start, leading to training difficulties and poor generalization (Silva & Gombolay, 2021). **(3) Challenge of synergistic updating of external knowledge and internal strategies.** Large Language Models (LLMs) have recently demonstrated impressive capabilities (Chowdhery et al., 2022; Brown et al., 2020), but constant changes in the external world can quickly render them obsolete for time-sensitive tasks (*e.g.*, news question answering). While some recent studies have attempted to directly map the rich semantic knowledge within LLMs to actions (Carta et al., 2023; Mezghani et al., 2023), these mappings cannot be updated or come with high update costs.

We propose using natural language as a knowledge source to bypass the difficulties of knowledge acquisition and representation. The language contains natural and flexible prior knowledge for efficient exploration and skill acquisition, enabling rapid generalization across tasks. Truly understanding the semantics and focusing on the key information is a good start to integrating external knowledge and internal strategies. To synergize the updating of external knowledge and internal strategies, external knowledge sources and internal strategies should retain the ability to learn and improve over time.

To bridge the gap between RL and human in handling sparse reward tasks, as shown in Figure 1, we propose a novel paradigm for guiding agent learning to absorb external knowledge called **PAE**: **P**lanner-**A**ctuator-**E**valuator. The Planner is equipped with a state-knowledge alignment mechanism. This mechanism enables the Planner to access external knowledge sources and retrieve suitable knowledge that aligns with the current state. This aligned knowledge is progressively provided to the Actor, increasing in complexity. The Actor leverages the state information and the external knowledge provided by the Planner for joint reasoning. It incorporates a cross-attention mechanism, allowing the Actor to precisely focus on the critical state and external knowledge features. Additionally, the Actor employs a discriminative network to reverse-infer the Planner's guidance, strengthening the connection between states and knowledge. The Evaluator calculates intrinsic rewards based on the quality of external knowledge the Planner provides and the Actor's reasoning effectiveness. These intrinsic rewards guide independent updates of the Planner and the Actor.

To summarize, our contributions are as follows:

- We present a novel framework, PAE, for learning to incorporate external knowledge to achieve better and faster solutions in sparse reward environments. We demonstrate that knowledge instructing significantly improves strategy exploration in RL.

- Our PAE framework introduces a circular feedback mechanism. The Evaluator's feedback keeps the Planner aware of the scenario. The external knowledge the Planner provides aligns with the Actor's current capabilities, enabling the Actor to master knowledge from easy to complex.

- Our proposed framework produces interpretable sequences of steps, with the Planner providing a comprehensible plan in natural language form.

We evaluated our proposed PAE framework across 11 challenging tasks from the BabyAI (Chevalier-Boisvert et al., 2018) and MiniHack (Samvelyan et al., 2021) environment suites. Compared to non-knowledge-guided baseline methods, our PAE framework improves performance by an average of 148% and achieves convergence at least 40% faster. Compared to baseline approaches using language guidance, the PAE framework exhibits an average performance improvement of at least 13% and converges at least 31% faster.

## 2 RELATED WORK

We build upon extensive work in areas related to the study of knowledge and reinforcement learning (RL). Our focus lies in knowledge acquisition and representation, the integration of external knowledge and internal strategies, and exploration in RL.

**Knowledge Acquisition and Representation** Converting and representing extracted knowledge effectively is the primary challenge in enhancing a reinforcement learning agent's capabilities with external knowledge. Imitation learning and inverse reinforcement learning capture domain knowledge by mimicking human actions via state-action pairs, yet these often require laborious human labeling and feedback (Codevilla et al., 2018; Ho & Ermon, 2016). Representing knowledge as a graph (Xu et al., 2020; Zhang et al., 2022), such as a knowledge or scene graph, encounters issues of limited structured data and sparse valid information. He et al. (2017) treat documents as external knowledge; the agent must learn to interpret them for task-solving. Bougie & Ichise (2018) use environmental data from object detectors as external knowledge, broadening its applicability. In contrast, our approach employs natural language directly as external knowledge, eliminating the need for preprocessing. This fosters agent exploration guided by human-intuitive, natural interaction.

**External Knowledge and Internal Strategy Integration** Incorporating external knowledge to enhance agent decision-making is crucial. Some methods teach AI to use language for querying external knowledge (Carta et al., 2022; Liu et al., 2022) but grapple with generating sizable, retrievable language queries. Several approaches (Humbird et al., 2018; Silva & Gombolay, 2021) aim to embed human domain knowledge into neural decision trees yet face the challenge of bridging tree-to-neural network gaps. Our focus is linking natural language with policies to guide agent learning. Like Language-conditioned RL, our approach trains agents to follow instructions in interactive environments (Luketina et al., 2019). While prior research explored this setting for various tasks in 2D or 3D environments (Colas et al., 2020; Chevalier-Boisvert et al., 2018), we aim to craft a comprehensive framework that encompasses knowledge selection and representation, the integration of external knowledge and internal strategies, and updating knowledge-guided strategy. Our approach absorbs and understands key parts of the language for joint reasoning and selects external knowledge to provide knowledge appropriate to the current agent's capabilities. This results in enhanced task performance, greater adaptability, and improved interpretability. Some more recent work (Carta et al., 2023; Du et al., 2023) have extracted and employed knowledge from LLMs for decision-making tasks. These approaches often directly map the internal knowledge of LLMs to actions. However, this mapping weakens the policy update potential of reinforcement learning algorithms and might present challenges regarding update costs for time-sensitive tasks.

**Exploration in RL** Exploration stands as a crucial challenge in reinforcement learning, covering $\epsilon$-greedy action selection (Strouse et al., 2021), state counting (Bellemare et al., 2016), curiosity-driven exploration (Schmidhuber, 1991), and intrinsic motivation (Oudeyer et al., 2007). Our approach aligns with intrinsic motivation, where novelty fosters access to new states. Other forms of intrinsic motivation include empowerment (Klyubin et al., 2005), promoting agent control over the environment, and goal diversity (Pong et al., 2020), encouraging increased entropy within the goal distribution. In contrast, we propose intrinsic motivation as a "cyclic feedback mechanism": The Evaluator's intrinsic reward prompts the Planner to equip the Actor with knowledge suitable for its current capacity, guiding the Actor to execute tasks aided by external knowledge. Our approach also emphasizes enhancing reinforcement learning through language-guided exploration (Mu et al., 2022; Carta et al., 2022). This methodology typically assumes an annotator's presence to generate language descriptions for intrinsic rewards, guiding exploration. We adopt this framework while automating and unifying the entire process, encompassing knowledge representation and selection, integrating external knowledge and intrinsic strategies, and computation of guided intrinsic rewards.

## 3 PRELIMINARIES

We formally define the problem formulation of Knowledge-Instructed Reinforcement Learning. We consider augmented Partially Observable Markov Decision Processes (POMDPs) (Kaelbling et al., 1998). Our augmented POMDP can be described by a 7-tuple $\mathcal{M} = \langle \mathcal{S}, \mathcal{A}, \mathcal{T}, \mathcal{R}, \Omega, O, \mathcal{K} \rangle$. $\mathcal{S}$ is the state space. $\mathcal{A}$ donates the action space. $\Omega$ is the observation space. $\mathcal{K}$ is the knowledge set. $\mathcal{T} : \mathcal{S} \times \mathcal{A} \times \mathcal{S} \mapsto \mathbb{R}^+$ is the state transition function. $\mathcal{O} : \mathcal{S} \times \mathcal{A} \times \Omega \mapsto \mathbb{R}^+$ is the observation

function. $\mathcal{R} : \mathcal{S} \times \mathcal{A} \times \mathcal{K} \mapsto \mathbb{R}$ represents the state-action reward. At each time step $t$, the agent receives an observation $o_t \in \Omega$ and a knowledge $k \in \mathcal{K}$. Based on these inputs, the agent selects an action $a_t \in \mathcal{A}$, and transits to the next state $s_{t+1} \in S$. Via RL, our goal is to find some policy $\pi : \mathcal{S} \times \mathcal{K} \mapsto \mathcal{A}$ that maximizes the expected return.

We assume that the environment provides knowledge following previous work (Jiang et al., 2019; Mirchandani et al., 2021; Waytowich et al., 2019; Mu et al., 2022). We impose no specific representation requirements and present all knowledge in natural language. Natural language's hierarchical structure and rich semantics profoundly impact an agent's cognitive functions (Colas et al., 2022). Agents can update internal strategies guided by language's generality and abstraction, enhancing comprehension of complex concepts and alignment with human values. Many modern RL environments include language by default (Mu et al., 2022), like BabyAI (Chevalier-Boisvert et al., 2018), NetHack (Küttler et al., 2020), MiniHack (Samvelyan et al., 2021), text-based games (Côté et al., 2019; Shridhar et al., 2020; Urbanek et al., 2019), and most video games. In language-absent environments, language descriptions relevant to the agent's state can be produced by generative models of language trained on gigantic amounts of text such as GPT (Brown et al., 2020), PaLM (Chowdhery et al., 2022), and Llama (Touvron et al., 2023).

## 4 METHOD

We introduce the PAE framework, a novel approach for instructing agents to learn to absorb knowledge. As shown in Figure 1, PAE consists of three core components: (1) the **Planner**, a policy network that delivers language-based external knowledge guidance, adapting in complexity following both task features and the Actor's current capabilities; (2) the **Actor**, a knowledge-conditioned policy, designed to integrate external knowledge and environment states for joint reasoning; and (3) the **Evaluator**, a module that offers intrinsic rewards to both the Planner and Actor, guiding their strategy updates.

### 4.1 ALIGNING EXTERNAL KNOWLEDGE AND STATES VIA PLANNER

To enhance guidance, the Planner must progressively select increasingly challenging knowledge for the Actor during the training procedure. This presents two key challenges: 1) correctly understanding knowledge in natural language and aligning it with the current environmental state, and 2) adjusting the complexity of the chosen knowledge based on the Actors' abilities. To achieve this, we consider the Planner as a policy network $\pi_p(k|s_0, \mathcal{K}; \boldsymbol{\theta})$ modeled by an MDP: it inputs an initial state $s_0$ [1] and a set of knowledge $\mathcal{K}$ and selects a piece of knowledge $k \in \mathcal{K}$ as output. The Planner provides new knowledge whenever a new episode starts, or the Actor reaches an intrinsic goal guided by the knowledge. The Planner receives rewards only when the Actor completes the task using the provided knowledge, aiming to maximize cumulative rewards. Figure 2 provides an overview of the Planner.

**Encoding.** To assist the Planner in better comprehending hidden information within the environment and knowledge, we introduce a state encoder and a knowledge encoder to encode state and knowledge information to embedded features separately. Specifically, for the state encoder, visual state embeddings $s_0 \in \mathbb{R}^{HW \times c}$ are fed to a shape-preserving 2D convolution layer to generate the state feature $\hat{s}_0 \in \mathbb{R}^{HW \times d_s}$. To capture both the content and spatial characteristics of the states, position embeddings $\mathbf{E}_{pos}$ are added to $\hat{s}_0$, which is formulated as:

$$\hat{\mathbf{s}}_0 = \text{Conv}(s_0) + \mathbf{E}_{pos}, \qquad s_0 \in \mathbb{R}^{HW \times C} \quad \hat{s}_0, \mathbf{E}_{pos} \in \mathbb{R}^{HW \times d_s}, \tag{1}$$

where $c$ is the dimension of embedding vectors and $d_s$ is the channels of feature maps after convolution. For the knowledge encoder, each unit of knowledge $k_i \in \mathcal{K}$ is encoded using a pre-trained language model (*e.g.* BERT-base) with frozen parameters, yielding the knowledge embeddings $\hat{k}_i$:

$$\hat{\mathbf{k}} = [\hat{k}^{(1)}, \hat{k}^{(2)}, \dots, \hat{k}^{(n)}] = \text{Proj}(\text{LM}([k^{(1)}, k^{(2)}, \dots, k^{(n)}])), \qquad \hat{\mathbf{k}} \in \mathbb{R}^{n \times d_k}, \tag{2}$$

where $\text{Proj}(\cdot)$ is a linear projection layer and $d_k$ is the dimension of knowledge embedding vectors.

---

[1] The reason we choose $s_0$ instead of $s_t$ is that the Planner is required to provide macro-level guidance based on its curriculum, rather than step-by-step guidance based on the Actor's specific actions at timestep $t$.

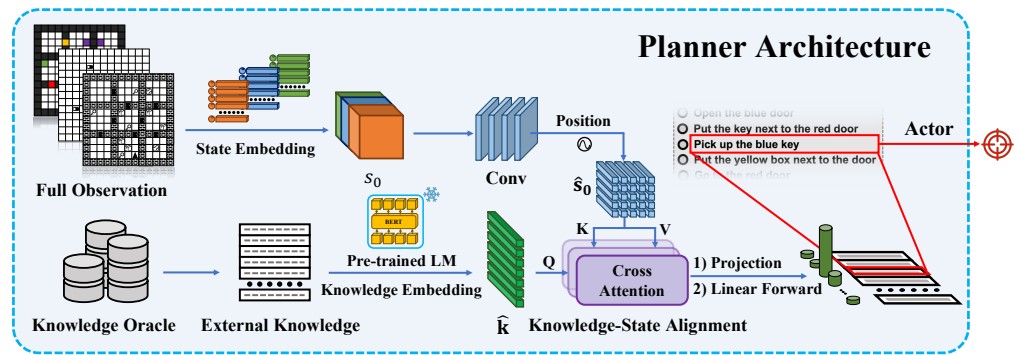

Figure 2: Overview of the Planner Network

**Alignment.** To better align external knowledge with the current environment, we utilize the scaled dot-product cross-attention mechanism to enable the Planner to attend between knowledge and states. Specifically, we compute the query $Q$ using the knowledge embeddings $\hat{\mathbf{k}}$ and compute the key $K$ and value $V$ using the position-embedded state feature $\hat{\mathbf{s}}_0$. In summary, the alignment procedure can be formulated as:

$$\hat{\mathbf{k}}_{s_0} = \text{Attention}(Q, K, V) = \text{SoftMax}\left(\frac{QK^T}{\sqrt{d_k}}\right)V$$

$$\text{where } Q = \hat{\mathbf{k}}W_Q, \ K = \hat{\mathbf{s}}_0 W_K, \ V = \hat{\mathbf{s}}_0 W_V,$$

(3)

where $W_Q \in \mathbb{R}^{d_k \times d_k}$, $W_K \in \mathbb{R}^{d_s \times d_k}$ and $W_V \in \mathbb{R}^{d_s \times d_k}$ are learnable projection matrices.

**Forwarding.** The cross-attention layer produces $n$ context vectors $\hat{\mathbf{k}}_{s_0} = [\hat{k}_{s_0}^{(1)}, \hat{k}_{s_0}^{(2)}, \dots, \hat{k}_{s_0}^{(n)}]$ that aggregates critical information from both the knowledge set and the state, emphasizing knowledge most relevant to the current state. Then, $\hat{\mathbf{k}}_{s_0} \in \mathbb{R}^{n \times d_k}$ will be linear projected to $n$ 1-dimensional scales and go through a softmax layer, aiding the Planner in selecting the appropriate instruction for the Actor.

$$\pi_p(k|s_0, \mathcal{K}; \boldsymbol{\theta}) = \text{SoftMax}(\text{LN}(\hat{\mathbf{k}}_{s_0})).$$

(4)

## 4.2 TRAINING ACTOR-SPECIFIC INCREMENTAL SKILLS WITH EXTERNAL KNOWLEDGE

The Actor is a knowledge-conditioned policy parameterized as $\pi_a(a_t|{}^p s_t, k; \boldsymbol{\omega})$. Once the Planner provides the Actor with suitable external knowledge, we aim for the Actor's competence in two key areas. First, like the Planner, the Actor must comprehensively grasp knowledge in its natural language form. Second, we want the guiding knowledge to influence the state reached by the Actor, such that different knowledge leads to distinct skills. Accordingly, we seek to strengthen the correlation between knowledge and state.

**Encoding, Alignment & Forwarding.** To master language comprehension and align knowledge with current observable states, the Actor adopts a three-stage pipeline similar to the Planner: encoding, alignment, and forward. However, unlike the Planner, the Actor receives ${}^p s_t \in \mathbb{R}^{hw \times c}$, a partially observable state of size $h \times w$, and $k$, a single piece of knowledge filtered by the Planner, at each time step $t$. In the Encoding stage, due to the reduced information content of the input state ${}^p s_t$, we remove the shape-preserving convolution and directly add position information to state embeddings. Meanwhile, we also utilize the parameter-frozen language model to encode the existing knowledge effectively:

$$\begin{aligned}
{}^p\hat{\mathbf{s}}_t &= {}^p s_t + \mathbf{e}_{pos} & {}^p\hat{\mathbf{s}}_t, {}^p s_t, \mathbf{e}_{pos} \in \mathbb{R}^{hw \times c} \\
\hat{k} &= \text{Proj}(\text{LM}(k)) & \hat{k} \in \mathbb{R}^{1 \times d_k}
\end{aligned}$$

(5)

During the Alignment stage, we also incorporate the scaled dot-product cross-attention mechanism to achieve the knowledge-state alignment.

$$^p\hat{k}_{s_t} = \text{Attention}(Q, K, V) = \text{SoftMax}\left(\frac{QK^T}{\sqrt{d_k}}\right)V$$

$$\text{where } Q = \hat{k}W_Q, \ K = {}^p\hat{\mathbf{s}}_t W_K, \ V = {}^p\hat{\mathbf{s}}_t W_V$$

(6)

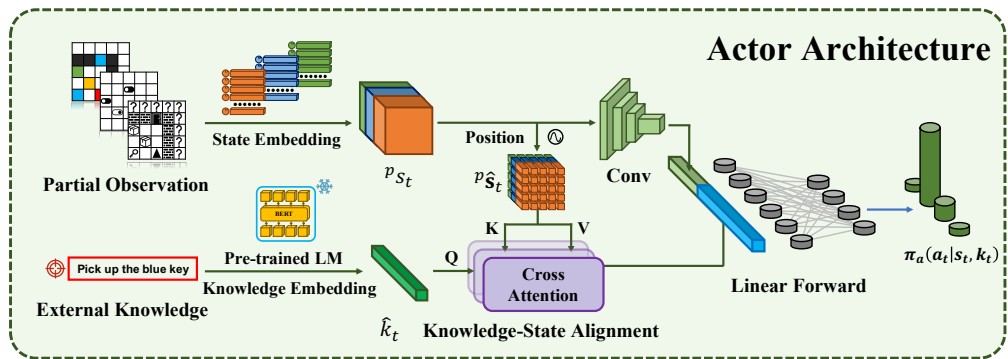

Figure 3: Overview of the Actor Network

Moving on to the Forward stage, we aim to enhance the Actor's attention to current state information. To achieve this, we employ strided convolution to extract state information further. We then concatenate this extracted information with the state-aligned knowledge, passing it through a Linear and a softmax layer to generate the distribution over actions.

$$\pi_a(a_t|s_t, k; \boldsymbol{\omega}) = \text{SoftMax}(\text{LN}([\text{Conv}(^p s_t), {}^p \hat{k}_{s_t}])). \tag{7}$$

**Maximize the mutual information between knowledge and states.** We utilize the mutual information maximization objective to build on the idea that knowledge should dictate the states the agent visits. Our approach maximizes the mutual information $I(s; k)$ between knowledge and states. The objective can be written as follows:

$$\begin{aligned} I(s; k) &= \mathcal{H}(k) - \mathcal{H}(k|s) \\ &= \mathbb{E}_{k \sim p(k), s \sim \pi}[\log p(k|s)] - \mathbb{E}_{k \sim p(k)}[\log p(k)] \\ &\geq \mathbb{E}_{k \sim p(k), s \sim \pi}[\log q_\phi(k|s) - \log p(k)]. \end{aligned} \tag{8}$$

As we cannot compute $p(k|s)$ exactly by integrating over all states and their associated knowledge, we approximate $p(k|s)$ using a trainable discriminative network $q_\phi(k|s)$. Previous work (Eysenbach et al., 2018) has proved that maximizing the mutual information between $s$ and $k$ can translate into maximizing the variational lower bound obtained by replacing $p(k|s)$ with $q_\phi(k|s)$ (as shown in Equation 8). From this variational lower bound, we give the mutual information rewards $r_{MI}$ used to encourage the Actor to strengthen the connection between state and knowledge:

$$r_I = \log q_\phi(k|^p s) - \log p(k). \tag{9}$$

The discriminative network $q_\phi(k|^p s)$ is optimized using a cross-entropy loss $\mathcal{L}_\phi$, denoting as:

$$\mathcal{L}_\phi(^p s, k) = -\sum_{k' \in \mathcal{K}} (k' = k) \cdot \log(q_\phi(k'|^p s)). \tag{10}$$

In summary, the Planner samples a piece of knowledge $k$ from the logits output by $\pi_p(k|s_0, \mathcal{K}; \boldsymbol{\theta})$ and provides it to the Actor. The Actor employs the conditional network $\pi_a(a_t|^p s_t, k; \boldsymbol{\omega})$ to make decisions in the environment guided by this knowledge $k$. The Actor is motivated to explore states that offer higher rewards and have a close relationship with $k$. The discriminative network $q_\phi(k|s)$ uses the $k$ provided by the Planner as ground truth and updates through supervised learning to better infer knowledge $k$ from the visited states. $p(k)$ is provided by the Planner's policy network $\pi_p(k|s_0, \mathcal{K}; \boldsymbol{\theta})$. More implementation details are shown in Appendix A.3. Algorithm 1 in Appendix A.4.1 summarizes our PAE procedure.

### 4.3 EVALUATION MECHANISM

Reinforcement learning in long-term and sparse reward tasks is often difficult and requires a lot of trial and error. The standard solution to speed up this process is introducing additional reward signals to better guide learning. Incorporating external knowledge offers a more straightforward approach to reward shaping.

For the Actor, the external knowledge from the Planner serves as the missing signal in the environment. In addition to the extrinsic reward for accomplishing the final task, the Actor receives an

additional intrinsic reward of $+1$ for each subtask completed per the Planner's guidance. Consequently, the total rewards for the Actor comprise three terms: environmental extrinsic rewards $r_{ex}$, the maximization of the mutual information term $r_{MI}$, and intrinsic rewards $r_{in}$:

$$r_{actor} = r_{ex} + r_{in}(\mathbb{I}(\pi_a(a_t|s_t, k; \boldsymbol{\omega}) \mapsto k)) + \alpha_I r_I, \qquad (11)$$

where $\mathbb{I}$ is the indicator function, $\pi_a(a_t|^p s_t, k; \boldsymbol{\omega}) \mapsto k$ denotes that the Actor completed the subtask following external knowledge guidance, $\alpha_I > 0$ is a scaling factor.

For the Planner, we follow the previous approach (Mu et al., 2022; Campero et al., 2020) of using the Actor's task completion as an intrinsic reward. This task is neither too easy (*i.e.*, the knowledge given by the Planner is too simple) nor impossible (*i.e.*, the knowledge given by the Planner is too complex) for the Actor to complete the current task, *i.e.*, an automatic curriculum. Specifically, we evaluate the Planner using the number of steps required for the Actor to follow the guidance proposed by the Planner; if the number of steps required for the Actor to follow the guidance is within the threshold $t^*$, or if the guidance can not be followed until the end of the episode, the Planner receives a reward of $-\beta$, otherwise, the Planner gets a reward of $+\alpha$:

$$r_{planner} = r_{ex} + r_{in}; \; r_{in} = \begin{cases} +\alpha & \text{if } t \geq t^* \\ -\beta & \text{if } t < t^* \text{ or } t > t_{max} \end{cases} \qquad (12)$$

For the value of $t^*$, we use an adaptive but heuristic approach, where the threshold increases linearly by one when the Actor successfully follows the guidance ten times.

## 5 EXPERIMENTAL EVALUATION

Our experimental evaluation aims to test our central hypothesis: that external knowledge improves the exploration efficiency for RL algorithms in sparse reward environments. We conducted a series of experiments organized as follows: 1) In Section 5.1, we quantitatively evaluate the exploratory capabilities of the PAE by comparing it to various baseline methods. 2) In Section 5.2, we revealed the underlying mechanisms of the PAE through ablation experiments. 3) In Section 5.3, we elucidate the interpretability of PAE's strategy by visualizing the agent's learning process at different stages. See Appendix A.5 for additional experimental results. More implementation details of PAE are shown in Appendix A.3.

**Environments**: We evaluated our method across two task types, totaling six environments within the BabyAI environment: Key Corridor tasks (KEYCORRS3R3, KEYCORRS4R3, KEYCORRS5R3) and Obstructed Maze tasks (OBSTRMAZE1DL, OBSTRMAZE2DLHB, OBSTRMAZE1Q). The suffix signifies the environment's size; a larger environment increases the exploration difficulty in the same type of task. To demonstrate PAE's scalability, we extended our PAE approach to the more challenging MiniHack tasks. MiniHack consists of procedurally generated tasks within a roguelike game, offering a richer observation space than the BabyAI environment. Additionally, it presents up to 75 dimensions of structured and context-sensitive action space. Our comparisons of PAE with baseline methods encompassed five MiniHack environments: LAVACROSS-RING, LAVACROSS-POTION, LAVACROSS-FULL, RIVER-MONSTER, and MULTIROOM-N4-MONSTER. See Appendix A.2 for more details.

We follow the previous work (Mu et al., 2022) to introduce knowledge. In BabyAI, the Planner receives a fully observed state and a set of 652 instructions from the BabyAI platform. Meanwhile, the Actor gets a $7 \times 7$ representation of its field of view and a single instruction filtered by the Planner. See Table 3 in Appendix A.2 for a list of knowledge provided by BabyAI. In MiniHack, the Planner and Actor have the same field of view. Observations consist of a $21 \times 79$ matrix of glyph identifiers, a 21-dimensional vector with the agent's statistics (like location and health), and a series of 256-character knowledge entries. The Planner accesses multiple knowledge entries, whereas the Actor receives a selectively filtered entry from the Planner. See Table 4 in Appendix A.2 for a list of knowledge provided by MiniHack.

### 5.1 QUALITATIVE RESULTS: COMPARE TO BASELINES

We compare PAE with six baseline algorithms across three categories to highlight PAE's advantages. 1) Vanilla Baseline Algorithms: **IMPALA** (Espeholt et al., 2018), a standard asynchronous

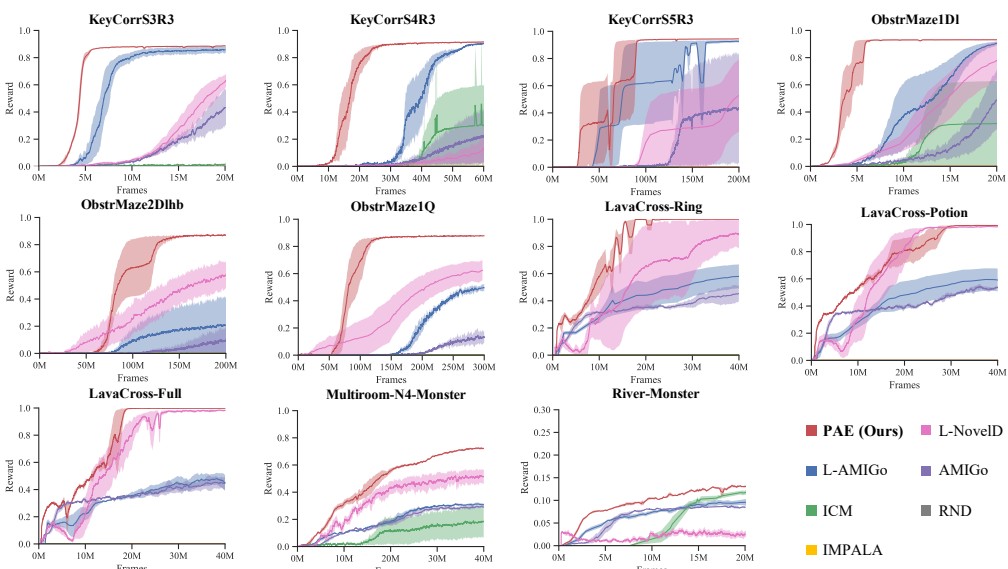

Figure 4: Performance of PAE, IMPALA, RND, ICM, AMIGo, L-AMIGo and L-NovelD in eleven environments, with error regions to indicate standard deviation over five random seeds.

actor-critic method, focuses on rapid parallel training and learns only from raw environmental rewards, without any intrinsic motivation or external guidance. 2) Intrinsic Motivation Algorithms: **RND** (Burda et al., 2018) uses a randomly initialized neural network to compute prediction errors, which serve as intrinsic rewards for exploring new states. **ICM** (Pathak et al., 2017) encourages exploration by predicting the outcomes of actions and generating intrinsic rewards from prediction errors. **AMIGo** (Campero et al., 2020) pairs a goal-generating teacher with a goal-conditioned student policy, enriching environmental rewards with intrinsic goals. 3) Language-Instructed Algorithms: **L-AMIGo** (Mu et al., 2022) builds on AMIGo's model, using language to spotlight relevant abstract concepts in the environment. **L-NovelD** (Mu et al., 2022) combines natural language with an intrinsic motivation approach to reward states described in natural language that transitions from low to high novelty. While all the above baselines, except the vanilla ones, incorporate some form of external guidance, L-AMIGo is the most similar to ours. However, it introduces language knowledge solely at the level of external guidance without focusing on its integration with internal strategies. The full hyperparameter sweep for PAE and all baselines is reported in Appendix A.3.4.

As shown in Figure 4, PAE achieves almost the fastest convergence across all eleven environments. Notably, AMIGo, RND, ICM, and IMPALA fail to converge within the current step limit. Compared to the best performance in non-knowledge-guided baseline methods (AMIGo, ICM, RND), our PAE framework improves performance by an average of 148% and achieves convergence at least 40% faster. Compared to baseline approaches using language guidance, the PAE framework exhibits an average performance improvement of 13% and converges at least 31% faster. Tables 6 and 7 in Appendix A.5 give the quantitative performance of all models in the BabyAI and MiniHack environments, respectively. Furthermore, we observe excellent training stability in PAE, especially after convergence.

## 5.2 ABLATION EXPERIMENT AND ANALYSIS

To better understand the advantages PAE brings through introducing external knowledge, this section compares the mechanisms for introducing external knowledge in removing or replacing implementations. **Full-Model** is the full version of PAE. In **w/o Curriculum**, the Planner attaches equal weight to all external knowledge. **w/o Planner** eliminates the entire Planner's guidance to the Actor. Table 1 displays the final mean extrinsic rewards and the number of steps (in millions) needed for each model to converge. Each entry consists of two rows of results, with the top row being the average extrinsic reward at the end of training and the bottom row being the minimal stable steps to attain that reward. Lower bottom row values signify quicker convergence, and "$> x$" indicates a lack of convergence within the maximum training steps "$x$". Training curves for the ablation study are provided in Appendix A.5.

Table 1: Comparison of PAE and ablation models.

| Model | Key Corridor Tasks | | | Obstructed Maze Tasks | | |
|---|---|---|---|---|---|---|
| | **KEYCORRS3R3** | **KEYCORRS4R3** | **KEYCORRS5R3** | **OBSTRMAZE1D1** | **OBSTRMAZE2D1HB** | **OBSTRMAZE1Q** |
| **Full-Model** | **0.89 ± 0.002** 
 **6M** | **0.92 ± 0.005** 
 **30M** | **0.94 ± 0.001** 
 **90M** | **0.93 ± 0.004** 
 **6M** | **0.87 ± 0.018** 
 **150M** | **0.89 ± 0.006** 
 **150M** |
| w/o Curriculum | 0.86 ± 0.020 
 20M | 0.00 ± 0.000 
 >60M | 0.00 ± 0.000 
 >200M | **0.93 ± 0.004** 
 12M | 0.71 ± 0.324 
 >200M | 0.85 ± 0.047 
 250M |
| w/o Planner | 0.00 ± 0.000 
 >20M | 0.00 ± 0.000 
 >60M | 0.00 ± 0.000 
 >200M | 0.00 ± 0.000 
 >20M | 0.00 ± 0.000 
 >200M | 0.00 ± 0.000 
 >300M |

We can see that the 'w/o Curriculum,' the Planner attaches equal weight to all external knowledge, performs well in four of the six environments (KEYCORRS3R3, OBSTRMAZE1DL, OBSTRMAZE2DLHB, OBSTRMAZE1Q) but requires at least 54% more training steps compared to the FULL-MODEL in these four environments. 'w/o Curriculum' earned zero rewards in the other two environments (KEYCORRS4R3, KEYCORRS5R3). We analyze this because tasks like ObstructedMaze require more robust exploration to unlock critical states, while tasks like KeyCorridor emphasize easy-to-difficult completion of the final task. The 'w/o Planner' model earned no external rewards in all six environments. The above results and analysis prove the critical role of introducing external knowledge, especially of suitable difficulty, for exploration in sparse reward environments.

## 5.3 INTERPRETABILITY

One advantage of PAE's introduction of language knowledge is that it can provide insights into developing an agent's abilities during training. We first show in Figure 5 (a) that PAE has a similar capability to the approach of the automatic curriculum: Planner generates an interpretable curriculum. In the *KeyCorridorS3R3* environment, the Planner first provides the Actor with easily achievable external knowledge (*open the door*) and gradually increases the difficulty as training progresses (*go to the door*, *pick up the key*), ultimately converging to the final goal (*pick up a ball*).

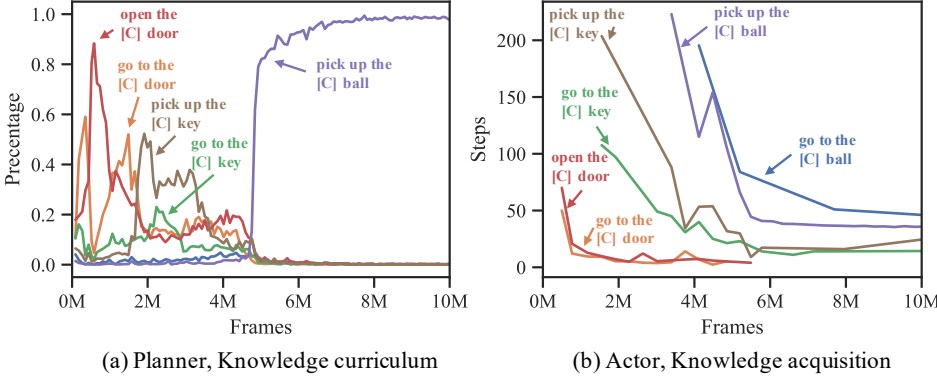

(a) Planner, Knowledge curriculum  (b) Actor, Knowledge acquisition

Figure 5: Interpretation of knowledge-instructed exploration. The knowledge curriculum introduced by the Planner (left) and the Actor's knowledge acquisition process (right) are illustrated in the *KeyCorridorS3R3* environment.

In addition to generating a similar curriculum, Figure 5 (b) also visualizes the number of steps required for the Actor to follow each external knowledge, offering insights into this incremental capability acquisition. We can see that the Planner provides increasingly difficult knowledge during the training process, reflected in a gradual increase in the probability of complex knowledge being presented. Conversely, the Actor gradually improves his proficiency in mastering each knowledge fragment during the training process, reflected in a gradual decrease in the required steps.

## 6 CONCLUSION

In this work, we propose Planner-Actor-Evaluator (PAE), a knowledge-instructed reinforcement learning framework for efficient exploration in sparse reward environments. PAE has several desirable properties: it achieves alignment and joint inference of external knowledge and internal agent states and emerges as an interpretable curriculum. Moreover, PAE is algorithm agnostic, making it compatible with any deep reinforcement learning algorithm.

ACKNOWLEDGEMENT

This work was partly supported by the Natural Science Foundation of China under Grant No. 62222606 and 62076238.

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

# A  APPENDIX

The supplementary material provides additional results, discussions, and implementation details.

- In Section A.1, we discuss the limitations of our PAE approach and provide insights on related work.
- In Section A.2, we detail the BabyAI and MiniHack environments and the chosen testing tasks.
- In Section A.3, we provide the implementation and training details for the PAE and the baseline algorithms.
- In Section A.4, we describe our PAE algorithm and the baseline algorithms.
- In Section A.5, we present additional experimental results and analysis.

## A.1  LIMITATIONS AND DISCUSSIONS

**Limitations** Despite the impressive results of our approach, we acknowledge at least two limitations. Firstly, our framework relies on environments like BabyAI and MiniHack to provide knowledge. This dependency means that applying PAE in environments lacking Oracle knowledge is challenging. However, we are fully committed to exploring the use of Large Language Models to provide knowledge. Secondly, our current approach only accepts unimodal knowledge in natural language. More exciting extensions will involve human interaction to provide more natural and multimodal knowledge, which deserves further exploration.

**Relationship with LLM-based frameworks** LLM-based agents are gaining popularity in solving RL tasks, and very recent studies have shown considerable potential (Carta et al., 2023; Chen et al., 2023). There are currently two main categories of LLM-based agents for decision-making: 1. Fine-tuning LLMs using RL for decision-making, and 2. Employing LLMs directly for decision-making as plug-ins. These differ significantly from PAE, which uses external knowledge to enhance RL algorithms. In contrast, recent LLM-based agent studies primarily showcase the abilities of LLMs or expand their capabilities using RL methods. In our PAE approach, LLMs are a tool to aid agents in understanding the semantics of knowledge.

## A.2  ENVIRONMENT AND TASK DETAILS

**BabyAI and Minigrid** The BabyAI platform enables research in grounded language learning involving humans. In BabyAI, agents maximize rewards by completing tasks within limited steps, guided by language instructions. The platform utilizes a grid world environment (Minigrid), a partially observable 2D grid world housing agents and objects (available in 6 colors): boxes, balls, doors, and keys. These entities occupy $N \times M$ tiled rooms interconnected by doors, which may be locked or closed. Agents can pick up, drop, and move objects, while doors require color-matching keys for unlocking. As shown in Figure 6, we evaluated our approach in the BabyAI across six environments spanning two task types: the Key Corridor task (KEYCORRS3R3, KEYCORRS4R3, KEYCORRS5R3) and the Obstructed Maze task (OBSTRMAZE1DL, OBSTRMAZE2DLHB, OBSTRMAZE1Q). Table 2 illustrates critical properties of the Minigrid environment. Table 3 shows the 66 categories of templates provided by BabyAI, totaling 652 pieces of language knowledge.

**Key Corridor** In the Key Corridor task, the 'S' in the environment name's suffix indicates the room's size, while 'R' signifies the number of rows. The agent must retrieve an item behind a locked door. The key, concealed in a different room, must be found by the agent exploring the environment. Figure 11 illustrates the flow chart for the agent to complete the Key Corridor tasks.

**Obstructed Maze** In the Obstructed Maze task, a small blue ball is concealed in a corner. With the door locked and blocked by the ball, the key to unlock it is hidden inside a box. The agent must execute a series of precise steps. Typically, this involves opening the box to obtain the correctly colored key, using the key to unlock the door, and subsequently accessing the door to reach the objective. Figure 12 illustrates the flow chart for the agent to complete the Obstructed Maze tasks.

**MiniHack** The MiniHack environment is a graphical adaptation of NetHack (Küttler et al., 2020), featuring a richer observation space compared to BabyAI. It includes more symbols and supports

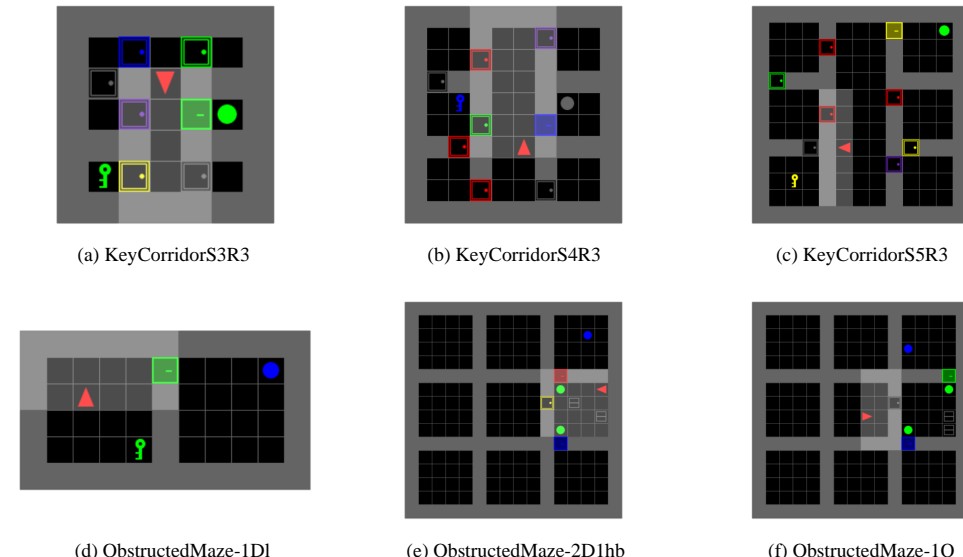

(a) KeyCorridorS3R3         (b) KeyCorridorS4R3         (c) KeyCorridorS5R3

(d) ObstructedMaze-1Dl      (e) ObstructedMaze-2D1hb      (f) ObstructedMaze-1Q

Figure 6: Six challenging environments in BabyAI we used to evaluate PAE.

up to 75 different actions. Observations comprise a $21 \times 79$ matrix of glyph identifiers, a 21-dimensional vector detailing agent statistics, location and health, and real natural language messages received during gameplay. As shown in Figure 7, we evaluate our approach in five environments of MiniHack, including two task types: navigation tasks (RIVER-MONSTER, MULTIROOM-N4-MONSTER) and skill acquisition tasks (LAVACROSS-RING, LAVACROSS-POTION, LAVACROSS-FULL). Navigation tasks in MiniHack challenge the agent program to navigate various obstacles, like crossing a river by maneuvering boulders or traversing intricate or randomly generated mazes to reach a specific destination. Skill Acquisition tasks leverage the extensive variety of NetHack's objects, monsters, and dungeon features, including their interplay. For instance, to wear a ring, the agent must choose the PUTON action, select the ring from the inventory, and decide which hand to wear it on. Table 2 details MiniHack's main features, while Table 4 enumerates some real natural language messages received by the agent during gameplay.

**Knowledge in BabyAI and MiniHack** (1) The knowledge presented in Table 3 for BabyAI and in Table 4 for MiniHack can be regarded as target states. For example, in MiniGrid, the message "go to the door" can be interpreted as "(in this state), go to the door." In MiniHack, the message "the o is killed!" can be interpreted as "(in this state), kill the o!". We can view these system text descriptions as the **required states** that the Planner hopes the Actor will achieve after taking a series of actions. Consequently, the Actor must either take the appropriate action to achieve these goals or extract useful strategies from the actions (trajectories) previously taken toward these goals. (2) Much of the knowledge in MiniHack does not directly contribute to task completion. Given that MiniHack's knowledge is derived from actual player feedback during gameplay, it naturally includes irrelevant or emotional messages, such as "ouch!" and "never mind." A rough estimate is provided here: an agent capable of solving the Lavacross task will encounter approximately 80 messages, of which only 6-10 (8-13%) are necessary for a successful trajectory. This requires the Planner to semantically understand these environment messages and filter and exclude those useless knowledge.

## A.3 IMPLEMENTATION DETAILS

### A.3.1 MODEL ARCHITECTURE DETAILS OF PLANNER

This section provides more details of the Planner architecture used in our method. Planner consists of three components, *i.e.*, encoding, alignment, and forwarding of states and knowledge.

**Encoding** Following BabyAI, the environment observation is a symbolic $H \times W \times 3$ fully observed representation, with $N$ varying according to the environment's size. Each cell in the $H \times W$ grid has 3 features indicating the object's type (*e.g.*, boxes, balls, doors, keys), color (from 6 possible

Table 2: Examples of BabyAI and MiniHack environments and their entity labeling.

| Illustration | Observation | Other Properties |
|---|---|---|
| 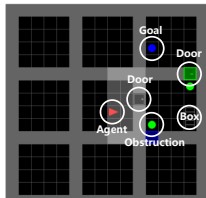 | **Observation**: In BabyAI, the agent's observation space is an egocentric 7x7 grid representation oriented in the direction the agent faces. Each cell within this grid has three defining features: object, color, and state. These features identify the object in the cell, its color, and its state (*e.g.*, distinguishing between a locked and unlocked door). | **Action**: Turn left, Turn right, Move forward, Pick up an object, Drop, Toggle, Done
**Reward**: A reward of '1 - 0.9 × (step_count / max_steps)' is given for success and 0 for failure.
**Termination**: The agent picks up the correct object.
Timeout. |
| 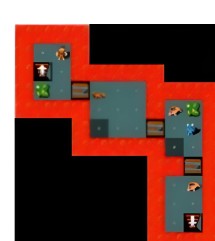 | **Observation**: In MiniHack, the agent's observation consists of a $21 \times 79$ matrix of glyph identifiers, a 21-dimensional vector with the agent's statistics (like location and health), and a series of messages. A "message" is the UTF-8 encoding of the on-screen message displayed at the top of the screen, represented as a 256-dimensional vector. Each glyph represents a completely unique entity, with integers ranging from 0 to MAX_GLYPH (5991). | **Action**: MiniHack contains 75 actions, including movement commands, managing inventory, casting spells, chopping wands, and more.
**Reward & Termination**: MiniHack's RewardManager allows you to specify one or more events that can yield varying (positive or negative) rewards. It also enables you to control which subsets of events are sufficient or required for episode termination. |

choices), and its state (like open or closed for doors). The Planner embeds these features into type/color/state embeddings with dimensions of 5, 3, and 2, resulting in a visual embedding $s_0$ of size $HW \times 10$. To process $s_0$, the Planner employs a 4-layer shape-preserving convolution neural network interleaved with the Exponential Linear Units, where each convolution layer has 16 filters with a size of $3 times 3$, a stride of 1 and padding of 1. The convolution neural network's output $\hat{s}_t$ provides an embedding layer sized $HW \times 16$. The Planner integrates a randomly initialized, trainable position embedding $\mathbf{E}_{pos}$ with $\text{Conv}(s_0)$ to encapsulate the state's content and spatial features and make preparations for alignment.

Meanwhile, the Planner utilized a pre-trained BERT model with frozen parameters to understand the semantics and encode knowledge. Specifically, we use the vector output of the encoder (in this case, BERT) in the [CLS] position for the sentence embedding. This model encoded a set of $n$ natural language instructions, supplied by the Oracle, into instruction embeddings with dimensions of $n \times 768$. Subsequently, a linear projection layer transformed these embeddings to produce outputs with dimensions of $n \times 16$, denoting as $\hat{\mathbf{k}}$.

**Alignment** To integrate the two types of input embeddings, $\hat{\mathbf{s}}_0 \in \mathbb{R}^{HW \times 10}, \hat{\mathbf{k}} \in \mathbb{R}^{n \times 16}$, we employ the scaled dot-product cross-attention mechanism. The query $Q$ is computed using $\hat{\mathbf{k}}W_Q$, and the key and value are derived from $\hat{\mathbf{s}}_t W_K$ and $\hat{\mathbf{s}}_t W_V$, respectively.

**Forward** After aligning knowledge and environment states, the cross-attention layer yields an output, $\hat{\mathbf{k}}_{s_0}$, which is a $n \times 16$ context vector capturing the correlation between the knowledge set and the current state. We project this context vector onto n-dimensional logits through a linear layer and finally get the most suitable external knowledge for the Actor after softmax sampling.

Table 3: All knowledge is provided by BabyAI and covers 66 template categories totaling 652 entries. [C] is one of 6 possible colors: green, grey, yellow, blue, purple, and red.

| **Go to** the <Object> | **Open** the <Object> | **Pick Up** the <Object> |
|---|---|---|
| **go to** the ball | | |
| **go to** the box | | **pick up** the ball |
| **go to** the door | **open** the box | **pick up** the box |
| **go to** the key | **open** the door | **pick up** the key |
| **go to** the <C> ball | **open** the <C> box | **pick up** the <C> ball |
| **go to** the <C> box | **open** the <C> door | **pick up** the <C> box |
| **go to** the <C> door | | **pick up** the <C> key |
| **go to** the <C> key | | |

| **Put the** <Object> **Next to the** <Object> | | |
|---|---|---|
| **put the** ball **next to** the ball | **put the** ball **next to** the <C> ball | |
| **put the** ball **next to** the box | **put the** ball **next to** the <C> box | |
| **put the** ball **next to** the door | **put the** ball **next to** the <C> door | **put the** <C> box **next to** the <C> ball |
| **put the** ball **next to** the key | **put the** ball **next to** the <C> key | **put the** <C> box **next to** the <C> box |
| **put the** box **next to** the ball | **put the** box **next to** the <C> ball | **put the** <C> box **next to** the <C> door |
| **put the** box **next to** the box | **put the** box **next to** the <C> box | **put the** <C> box **next to** the <C> key |
| **put the** box **next to** the door | **put the** box **next to** the <C> door | **put the** <C> key **next to** the <C> ball |
| **put the** box **next to** the key | **put the** box **next to** the <C> key | **put the** <C> key **next to** the <C> box |
| **put the** key **next to** the ball | **put the** key **next to** the <C> ball | **put the** <C> key **next to** the <C> door |
| **put the** key **next to** the box | **put the** key **next to** the <C> box | **put the** <C> key **next to** the <C> key |
| **put the** key **next to** the door | **put the** key **next to** the <C> door | |
| **put the** key **next to** the key | **put the** key **next to** the <C> key | |

(a) LavaCross-Ring  (b) LavaCross-Potion  (c) LavaCross-Full

(d) MultiRoom-N4-Monster  (e) River-Monster

Figure 7: Five challenging environments in MiniHack we used to evaluate PAE.

### A.3.2 MODEL ARCHITECTURE DETAILS OF ACTOR

This section details the Actor architecture used in our approach. The Actor consists of a policy network and a discriminative network.

The policy network adopts a similar network architecture as the Planner, with the difference that the inputs to the policy network at each time step $t$ are a partial observation of size $7 \times 7 \times 3$ and a single piece of knowledge $k$ filtered by the Planner. Due to the reduction of the information content of the input state, we remove the shape-preserving convolution and add the position information directly to the state embedding $^p s_t$, obtaining $^p \hat{s}_t$ of size $(7 \times 7) \times 10$. For the knowledge selected by the Planner, we utilize the same parameter-frozen BERT model to get the knowledge embedding $\hat{k}$ of dimension 64 after a linear projection. We compute the cross attention of $^p \hat{s}_t$ and $\hat{k}$ by using

Table 4: Some natural language messages in MiniHack. A "message" is the UTF-8 encoding of the on-screen message displayed at the top of the screen.

| River-Monster | MultiRoom-N4-Monster | LavaCross |
|---|---|---|
| -the stairs are solidly fixed to the floor.
-with great effort you move the boulder.
-that was close.
-it's solid stone.
-you don't have anything to zap.
-you hit the o.
-you try to crawl out of the water.
-your movements are slowed slightly because of your load.
-the o is killed!
-you try to move the boulder but in vain.
-you pull free from the o.
-you are almost hit by a dart.
-the o picks up a food ration.
-you have a little trouble lifing i –a scroll.
-you have a little trouble lifting j –a o corpse.
-… | -in what direction?
-that hurts!
-it's a wall.
-you see no door there.
-you kick at empty space.
-you strain a muscle.
-this door is locked.
-dumb move!
-ouch!
-as you kick the door, it crashes open!
-this door is broken.
-you hear the wailing of the banshee…
- as you kick the door, it shatters to pieces!
-this doorway has no door.
-what a strange direction!
-your leg feels better.
-never mind. | -what do you want to throw?
-you would burn to a crisp trying to pick things up.
-what do you want to use or apply?
-you move over some lava.
-the flint stone falls down the stairs.
-the flint stone hits another object.
-you see here a uncursed flint stone.
-sorry, i don't know how to use that.
-you float in the opposite direction.
-there is a staircase up here.
-you don't have anything to drink.
-in what direction?
-it's a wall.
-there is nothing here to pick up.
-you don't have anything to use or apply.
-emile's ghost touches you!
-agent's ghost touches you!
-… |

$\hat{k}_t$ to generate query $Q$, ${}^p\hat{s}_t$ for key $K$ and value $Q$. Then, we used four strided convolution layers to extract the state information to get $1 \times 1 \times 32$ state embedding. Finally, we concatenated state embedding and knowledge-state alignment embedding and passed it through one linear layer and softmax layer to get the logit distribution over the seven actions.

The Actor uses a separate discriminative network $q_\phi(k|{}^p s_t)$ to infer the knowledge provided by the Planner from the learned strategies. Intuitively, the Actor is motivated to bridge the connection between knowledge and states and explore states with a close relationship with $k$. The discriminative network $q_\phi(k|{}^p s_t)$ is optimized using a cross-entropy loss $\mathcal{L}_\phi$, denoting as:

$$\mathcal{L}_\phi({}^p s_t, k) = -\sum_{k' \in \mathcal{K}} (k' = k) \cdot \log(q_\phi(k'|{}^p s_t)) \tag{13}$$

Specifically, we employ a same-structure strided convolution network in the policy network to get the state feature ${}^p \tilde{s}_t$, and knowledge embeddings $\hat{\mathbf{k}}$ similar to the Planner:

$$\begin{aligned} \tilde{s}_t &= \mathrm{Conv}({}^p s_t) & &\in \mathbb{R}^{1 \times 32} \\ \hat{\mathbf{k}}_a &= \mathrm{Proj}(\mathrm{LM}([k^{(1)}, k^{(2)}, \ldots, k^{(n)}])) & &\in \mathbb{R}^{n \times 32} \end{aligned} \tag{14}$$

Then a dot-product operation between ${}^p \tilde{s}_t$ and knowledge embeddings is performed, followed by a softmax layer outputting $q_\phi(k|{}^p s_t)$:

$$q_\phi(k|{}^p s_t) = \mathrm{SoftMax}(\hat{\mathbf{k}}_a \cdot {}^p\tilde{s}_t{}^T) \tag{15}$$

Interestingly, whereas the discriminative network was initially derived from the definition of mutual information, we find that the discriminative network can be viewed as predicting and modeling the knowledge distribution of the planner from its perspective to strengthen the connection between state and knowledge.

### A.3.3 TRAINING DETAILS

Each model was trained using five independent seeds on a system with 112 Intel® Xeon® Platinum 8280 cores and 6 Nvidia RTX 3090 GPUs. Run times ranged from 10 hours (for OBSTRUCTED-MAZE1DL) to 100 hours (for the longest KEYCORRIDORS5R3 task).

### A.3.4 HYPERPARAMETERS

For PAE, we ran a grid search over batch size $\in$ $\{8, 32, 150\}$, unroll length $\in$ $\{20, 40, 100, 200\}$, entropy cost for the Actor $\in \{0.0001, 0.0005, 0.001\}$, the Actor learning rate $\in \{0.0001, 0.0005, 0.001\}$, the Planner learning rate $\in \{0.0001, 0.0005, 0.001\}$, entropy cost for the Planner $\in \{0.001, 0.005, 0.01\}$. Table 5 shows the best parameters obtained from the search.

For RND and ICM, we followed previous work (Raileanu & Rocktäschel, 2020) and used batch size of 32, unroll length of 100, RMSProp optimizer learning rate of 0.0001 with $\epsilon = 0.01$ and momentum of 0, intrinsic reward coefficient of 0.1, and entropy coefficient of 0.0005, which were the best values they found using grid searches in the same tasks.

For AMIGo and L-AMIGo, we followed previous work (Campero et al., 2020; Mu et al., 2022) and used teacher policy batch size of 32, student policy batch size of 32, teacher grounder batch size of 100, RMSProp optimizer learning rate of 0.0001 with $\epsilon = 0.01$ and momentum of 0, intrinsic reward coefficient of 1, unroll length of 100, value loss cost of 0.5, entropy cost of 0.0005, which were the best values for the same tasks described in their paper.

Table 5: Hyperparameters of PAE adopted in all the experiments.

| Parameter | Value |
|---|---|
| Actor batch size $B$ | 32 |
| Planner batch size $B_p$ | 32 |
| Unroll Length | 100 |
| Discount | 0.99 |
| Value loss cost | 0.5 |
| Actor entropy cost | 0.0005 |
| Planner entropy cost | 0.5 |
| Actor mutual information cost $\alpha_I$ | 0.0001 |
| Actor cross attention head number | 1 |
| Planner cross attention head number | 1 |
| Pretrained language model | Bert-base |
| Pretrained language model output dim | 768 |
| Actor language embedding dim | 64 |
| Planner language embedding dim | 256 |
| Planner embedding projection dim | 16 |
| Actor learning rate | 5e-4 |
| Planner learning rate | 5e-4 |
| Adam betas | (0.9, 0.999) |
| Adam $\epsilon$ | 1e-8 |
| Clip gradient norm | 40.0 |

## A.4 ALGORITHMS DETAILS

### A.4.1 ALGORITHMS DETAILS OF THE PAE

Algorithm 1 illustrates the overall rollout process of PAE. We implemented the PAE and reproduction baseline algorithms using TorchBeast[2] (Küttler et al., 2019), an IMPALA-based PyTorch platform for rapid asynchronous parallel training in reinforcement learning.

---

**Algorithm 1:** Asynchronous learning algorithm for PAE

---

**Input:** Buffer $\mathbf{B}$, Actor batch $\mathbb{B}$, Planner batch $\mathbb{B}_p$, Actor batch size $B$, Planner batch size $B_p$
**Initialize:** $\mathbf{B} \leftarrow \varnothing, \mathbb{B} \leftarrow \varnothing, \mathbb{B}_p \leftarrow \varnothing, \pi_a(a_t|s_t, k; \boldsymbol{\omega}) \leftarrow \boldsymbol{\omega}_0, \pi_p(k|s_0, \mathcal{K}; \boldsymbol{\theta}) \leftarrow \boldsymbol{\theta}_0$

1  **Function** INTERACT(Actor Policy: $\pi_a^*$, Planner Policy: $\pi_p^*$, Buffer: $\mathbf{B}$)**:**
2     $\{s_0, {}^p s_0\} \leftarrow$ Env.Reset(0)   ▷ $s_0$ denotes full observation, ${}^p s_0$ denotes partial observation
3     $k \leftarrow \pi_p^*(k_0|s_0, \mathcal{K}; \boldsymbol{\theta^*})$
4     $k_{steps} \leftarrow 0$
5     **while** True **do**
6        $a_t \leftarrow \pi_a^*(a_t|{}^p s_t, k; \boldsymbol{\omega^*})$
7        $\{s_{t+1}, {}^p s_{t+1}, r_t, k_{done}, s_{done}\} \leftarrow$ Env.Step($a_t$)   ▷ Interact with the environment
8        $k_{steps} \leftarrow k_{steps} + 1$
9        **if** $s_{done}$ **then**
10          $\{s_0, {}^p s_0\} \leftarrow$ Env.Reset(0)   ▷ This episode is finished
11          $k \leftarrow \pi_p^*(k_0|s_0, \mathcal{K}; \boldsymbol{\theta^*})$
12          $k_{steps} \leftarrow 0$
13       **end**
14       **if** $k_{done}$ **then**
15          $k \leftarrow \pi_p^*(k|s_0, \mathcal{K}; \boldsymbol{\theta^*})$   ▷ The Actor successfully followed the Planner's guidance
16          $k_{steps} \leftarrow 0$
17       **end**
18       Update $\mathbf{B}$   ▷ Update Buffer $\mathbf{B}$ using newly collected experience
19    **end**
20 **end**

21 **Function** MAIN()**:**
22    $\pi_a^* \leftarrow \pi_a.copy()$   ▷ Make a parameter-fixed copy of $\pi_a$ for environment interacting
23    $\pi_p^* \leftarrow \pi_p.copy()$   ▷ Make a parameter-fixed copy of $\pi_p$ for environment interacting
24    $Th \leftarrow$ Thread(INTERACT, $\pi_a^*, \pi_p^*, \mathbf{B}$)   ▷ Create a thread for environment interacting
25    **while** not converged **do**
26       Update $\mathbb{B} \leftarrow \mathbf{B}$   ▷ Sample batch $\mathbb{B}$ of size $B$ from Buffer $\mathbf{B}$
27       Train $\pi_a(a_t|{}^p s_t, k; \boldsymbol{\omega})$ on $\mathbb{B}$   ▷ Training policy $\pi_a$ using reinforcement learning
28       Train $q_\phi(k|{}^p s_t)$ on $\mathbb{B}$   ▷ Training discriminative network $q_\phi$ using supervised learning
29       Update $\pi_a^* \leftarrow \pi_a.copy()$   ▷ Update $\pi_a^*$ with latest model $\pi_a$
30       Update $\mathbb{B}_p$ with $\{s_0, k, k_{steps}, s_{done}\}$ from $\mathbb{B}_p$
31       **if** $|\mathbb{B}_p| > B_p$ **then**
32          Train $\pi_p(k|s_0, \mathcal{K}; \boldsymbol{\theta})$ on $\mathbb{B}_p$   ▷ Training policy $\pi_p$ using reinforcement learning
33          $\mathbb{B}_p \leftarrow \varnothing$
34          Update $\pi_p^* \leftarrow \pi_p.copy()$   ▷ Update $\pi_p^*$ with latest model $\pi_p$
35       **end**
36    **end**
37    $Th.join()$   ▷ Stop interacting with the environment
38 **end**

---

A.4.2   DETAILS OF THE BASELINE ALGORITHMS

Below, we describe the implementation details of the various baseline algorithms. **IMPALA** is an off-policy actor-critic framework. The actor-critic agent maintains a policy $\pi_\theta(a|x)$ and a value function $v_\theta(x)$, both parameterized by $\theta$. Both the policy and value functions are refined using the actor-critic update rule. IMPALA also incorporates an entropy regularization loss. The update is derived from the gradient of a specific pseudo-loss function:

$$\mathcal{L}_{\text{Value}}(\theta) = \sum_{s \in \text{T}} (v_s - V_\theta(x_s))^2$$

$$\mathcal{L}_{\text{Policy}}(\theta) = -\sum_{s \in \text{T}} \rho_s \log \pi_\theta(a_s|x_s)(r_s + \gamma v_{s+1} - V_\theta(x_s))$$

$$\mathcal{L}_{\text{Entropy}}(\theta) = -\sum_{s \in \text{T}} \sum_a \pi_\theta(a|x_s) \log \pi_\theta(a|x_s) \tag{16}$$

$$\mathcal{L}(\theta) = g_v \mathcal{L}_{\text{Value}}(\theta) + g_p \mathcal{L}_{\text{Policy}}(\theta) + g_e \mathcal{L}_{\text{Entropy}}(\theta).$$

This decoupled architecture facilitates high throughput. Nonetheless, there can be a bias since the policy generating the trajectory might lag several updates behind the learner's policy when computing the gradient. IMPALA addresses this potential bias by employing the V-trace, which balances the variance-contraction trade-off in these off-policy updates.

**RND** The main idea of RND is to calculate intrinsic reward based on the prediction error linked to an agent's transfer. RND employs two neural networks: a fixed, randomly initialized target network that defines the prediction task and a predictor network trained on the agent's collected data. RND considers the prediction error from the predictor network, based on features produced by the target network, as a reward for novel states. We followed AMIGo (Campero et al., 2020), using a re-implemented version based on TorchBeast (Küttler et al., 2019) provided by (Raileanu & Rocktäschel, 2020).

**ICM** utilizes curiosity as an intrinsic reward, defining it as the agent's error in predicting its own behavior's consequences within a feature space learned through a self-supervised inverse dynamics model. Specifically, ICM encodes states $s_t$ and $s_{t+1}$ as features $\phi(s_t)$ and $\phi(s_{t+1})$, respectively, trained to predict $a_t$ (*i.e.*, the inverse dynamics model). The forward model takes $\phi(s_t)$ and $a_t$ as inputs and predicts the feature representation $\phi(s_{t+1})$ for $s_{t+1}$. The prediction error within the feature space serves as a curiosity-driven intrinsic reward. We followed AMIGo (Campero et al., 2020) using a re-implemented version of ICM provided by (Raileanu & Rocktäschel, 2020). based on TorchBeast (Küttler et al., 2019).

**AMIGo** is a meta-learning method that automatically learns to self-propose adversarial motivational intrinsic goals. AMIGo consists of two subsystems: a goal-conditioned student policy that outputs the agent's actions in the environment and a goal-generating teacher that guides the student's training. These two components train in adversarial training, where the student maximizes rewards by reaching the goal as quickly as possible, and the teacher maximizes rewards by proposing goals that the student can reach but not too quickly. We use the open-source AMIGo (Campero et al., 2020) code[3] as a baseline for our PAE method.

**L-AMIGo** utilizes language as a generalized medium to emphasize relevant environmental abstractions. It is an extension of AMIGo where the teacher presents goals using language instead of coordinates $(x, y)$. The student is a conditional policy, receiving language goals $g_t$ instead of $(x, y)$. Specifically, the student network is decomposed into policy and grounding networks. The policy network produces a distribution of goals based on the student's state, while the grounding network predicts the probability of achieving the goal.

**L-NovelD** combines natural language with an intrinsic motivation approach to reward states described in natural language that transitions from low to high novelty. It is an extension of NovelD, which includes two main terms. The first term measures the novelty of state transitions, encouraging exploration without penalizing revisiting less novel states. The second term rewards the agent only for first-time state visits in an episode.

---

[3]https://github.com/facebookresearch/adversarially-motivated-intrinsic-goals

## A.5 Additional Results

### A.5.1 Quantitative Results

Figure 4 shows the training curves of our PAE method alongside the IMPALA, RND, ICM, AMIGo, L-NovelD, and L-AMIGo baseline algorithms. Each model was trained using five independent random seeds. The results reported for each baseline and each environment are the best performance configuration of the policy for that environment. Tables 6 and 7 give the quantitative results of these comparison experiments in the BabyAI and MiniHack environments, respectively, and show the average extrinsic reward and the number of steps (in millions) required for each model to converge. Each entry consists of two rows of results, with the top row being the average extrinsic reward at the end of training and the bottom row being the minimal stable steps to attain that reward. Lower bottom row values signify quicker convergence, and "$> x$" indicates a lack of convergence within the maximum training steps "$x$".

Table 6: Comparison of PAE and baseline methods in BabyAI environment

.

| Model | Key Corridor Tasks | | | Obstructed Maze Tasks | | |
|---|---|---|---|---|---|---|
| | KeyCorrS3R3 | KeyCorrS4R3 | KeyCorrS5R3 | ObstrMaze1D1 | ObstrMaze2D1hb | ObstrMaze1Q |
| **PAE (Ours)** | **0.89 ± 0.002** | **0.92 ± 0.005** | **0.94 ± 0.001** | **0.93 ± 0.004** | **0.87 ± 0.018** | **0.89 ± 0.006** |
| | **6M** | **30M** | **90M** | **6M** | **150M** | **150M** |
| L-AMIGo | 0.86 ± 0.040 | 0.90 ± 0.011 | 0.93 ± 0.016 | 0.90 ± 0.030 | 0.23 ± 0.403 | 0.47 ± 0.029 |
| | 12M | 60M | 160M | 20M | >200M | >300M |
| L-NovelD | 0.63 ± 0.097 | 0.15 ± 0.195 | 0.55 ± 0.484 | 0.79 ± 0.255 | 0.57 ± 0.164 | 0.60 ± 0.107 |
| | >20M | >60M | >200M | >20M | >200M | >300M |
| AMIGo | 0.43 ± 0.246 | 0.22 ± 0.343 | 0.43 ± 0.599 | 0.45 ± 0.366 | 0.10 ± 0.175 | 0.18 ± 0.023 |
| | >20M | >60M | >200M | >20M | >200M | >300M |
| ICM | 0.04 ± 0.052 | 0.30 ± 0.511 | 0.00 ± 0.000 | 0.31 ± 0.542 | 0.00 ± 0.000 | 0.00 ± 0.000 |
| | >20M | >60M | >200M | 15M | >200M | >300M |
| IMPALA | 0.00 ± 0.000 | 0.00 ± 0.000 | 0.00 ± 0.000 | 0.00 ± 0.006 | 0.00 ± 0.000 | 0.00 ± 0.000 |
| | >20M | >60M | >200M | >20M | >200M | >300M |
| RND | 0.00 ± 0.000 | 0.00 ± 0.000 | 0.00 ± 0.000 | 0.00 ± 0.000 | 0.00 ± 0.000 | 0.00 ± 0.000 |
| | >20M | >60M | >200M | >20M | >200M | >300M |

Table 7: Comparison of PAE and baseline methods in MiniHack environment

.

| Model | LavaCross-Ring | LavaCross-Potion | LavaCross-Full | Multiroom-N4-Monster | River-Monster |
|---|---|---|---|---|---|
| **PAE (Ours)** | **1.00 ± 0.001** | **0.99 ± 0.002** | **1.00 ± 0.001** | **0.72 ± 0.020** | **0.13 ± 0.018** |
| | **22M** | 35M | **24M** | >40M | >20M |
| L-AMIGo | 0.57 ± 0.223 | 0.58 ± 0.214 | 0.45 ± 0.111 | 0.31 ± 0.035 | 0.09 ± 0.016 |
| | >40M | >40M | >40M | >40M | >20M |
| L-NovelD | 0.88 ± 0.197 | **0.99 ± 0.009** | 0.98 ± 0.015 | 0.51 ± 0.122 | 0.02 ± 0.017 |
| | >40M | **30M** | 30M | >40M | >20M |
| AMIGo | 0.46 ± 0.116 | 0.54 ± 0.048 | 0.44 ± 0.101 | 0.29 ± 0.026 | 0.08 ± 0.009 |
| | >40M | >40M | >40M | >40M | >20M |
| ICM | 0.00 ± 0.000 | 0.00 ± 0.000 | 0.00 ± 0.000 | 0.18 ± 0.199 | 0.12 ± 0.028 |
| | >40M | >40M | >40M | >40M | >20M |
| IMPALA | 0.00 ± 0.000 | 0.00 ± 0.000 | 0.00 ± 0.000 | 0.00 ± 0.000 | 0.00 ± 0.000 |
| | >40M | >40M | >40M | >40M | >20M |
| RND | 0.00 ± 0.000 | 0.00 ± 0.000 | 0.00 ± 0.000 | 0.00 ± 0.000 | 0.00 ± 0.000 |
| | >40M | >40M | >40M | >40M | >20M |

### A.5.2 Ablation Experiential Results

Figure 8 illustrates the effect of introducing external knowledge on PAE performance. **Full-Model** is the full version of PAE. **w/o Curriculum** provides randomized guidance to the Actors through external knowledge, while **w/o Planner** eliminates the Planner's guidance.

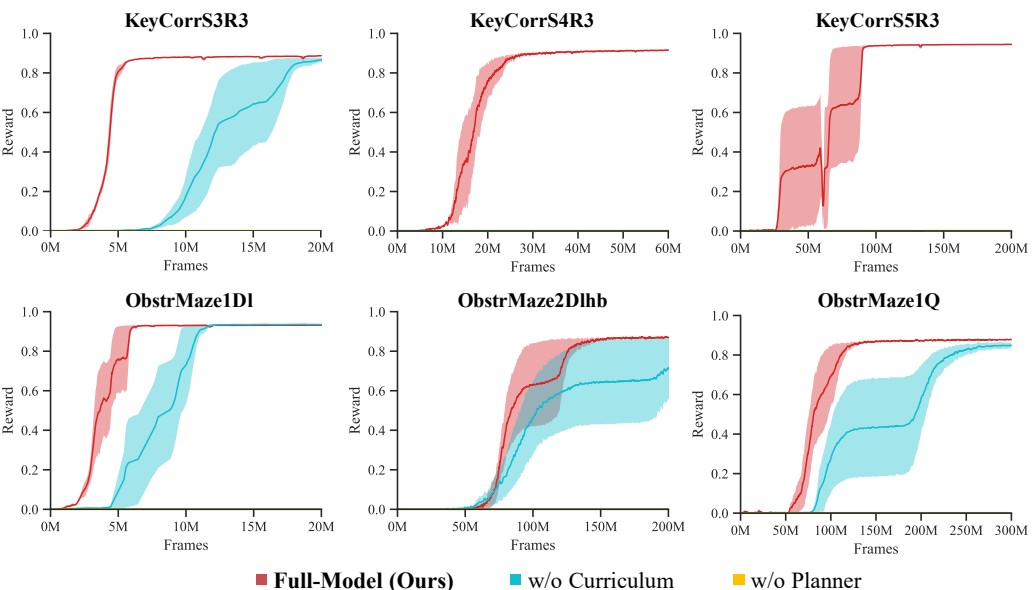

Figure 8: Ablation of PAE's external knowledge.

### A.5.3 QUALITATIVE RESULTS

Figure 9 shows an example of PAE completing a task in KEYCORRS5R3. In this example, the Actor first needs to explore the environment to find the key, then open the door of the corresponding color, drop the key, and retrieve the item behind the door. Figure 10 shows the flow of the Planner forming an automatic curriculum in *KeyCorrS5R3* and the progressive acquisition of skills by the Actor. Typically, in the early stages, the Planner provides easy guidance. As the training progresses and the Actor's ability increases, the Planner provides more hard guidance. We can see in the trajectory that the Planner and Actor are progressing together and eventually complete the task successfully.

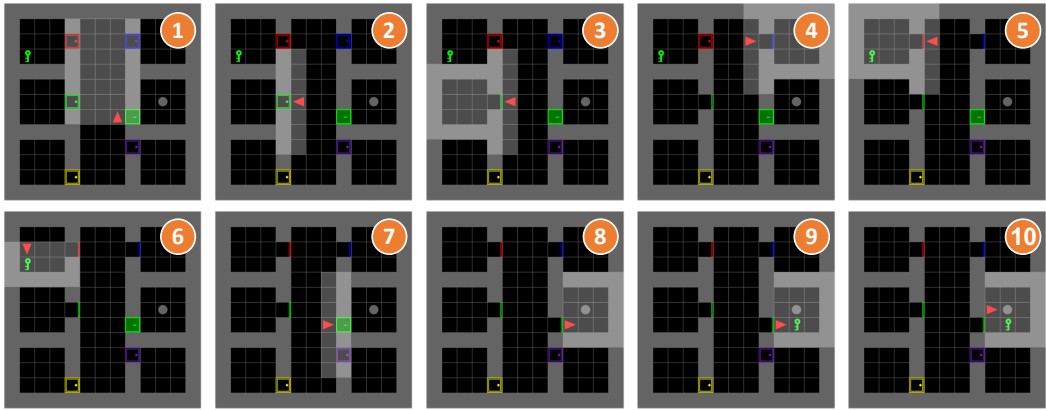

Figure 9: Example of PAE in completing a task in KEYCORRS5R3. To save space, we only sampled one sub-trajectory for display.

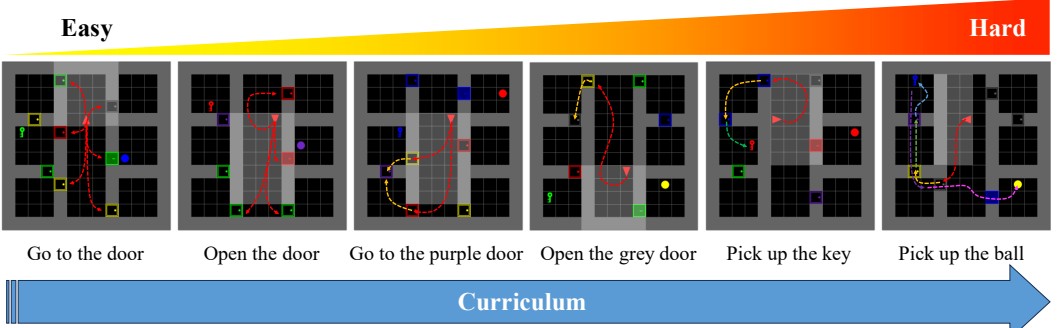

Figure 10: An illustration of the training process. The Planner forms an automatic curriculum as the training progresses, and the Actor progressively masters the skills.

## A.6 FLOW CHART

Figure 11 and Figure 12 show flowcharts of the agent completing the Key Corridor task and the Obstructed Maze task, respectively.

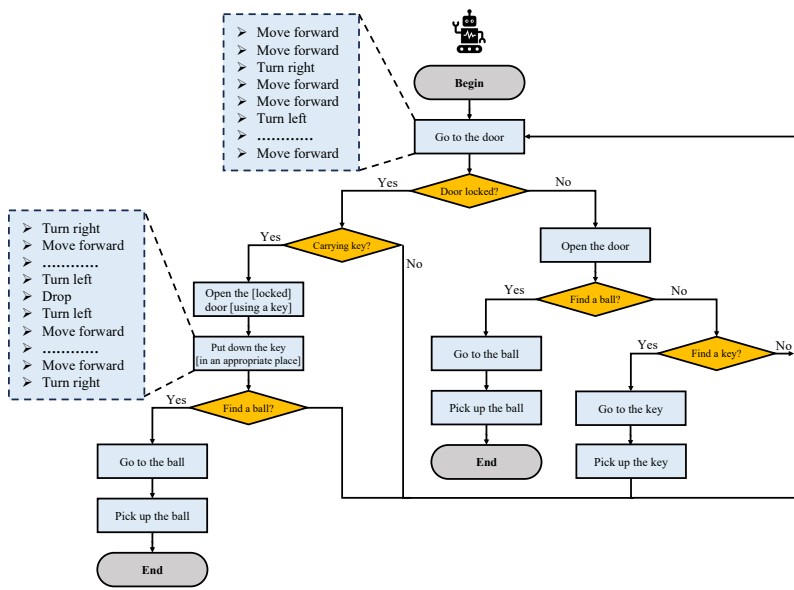

Figure 11: Flow chart of an agent completing Key Corridor tasks.

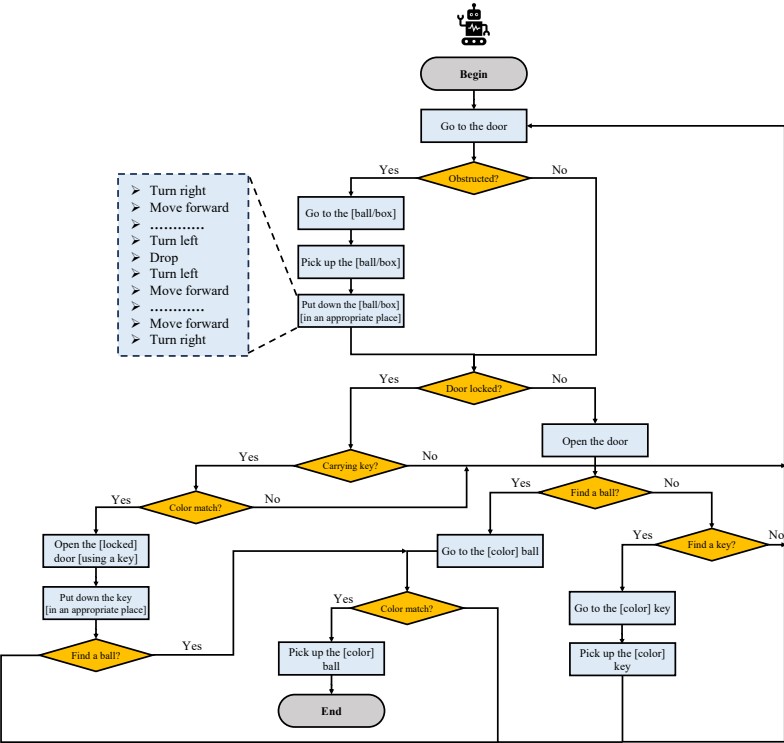

Figure 12: Flow chart of an agent completing Obstructed Maze tasks.

