# OpenReview forum: "PAE: Reinforcement Learning from External Knowledge for Efficient Exploration"
_ICLR.cc/2024/Conference — ICLR 2024 poster_

### Official Review · Reviewer_umQ7 · 2023-10-29

**Soundness:** 2 fair
**Presentation:** 3 good
**Contribution:** 2 fair
**Rating:** 6
**Confidence:** 3

**Summary:**

This paper presents a novel exploration framework for sparse reward environments, called Planner-Actor-Evaluator (PAE), which teaches RL agents to learn to absorb external knowledge. In particular, PAE adopts a state-knowledge alignment mechanism to enable a Planner to access external knowledge sources and retrieve suitable knowledge that aligns with the current environmental state. Then the actor leverages both of the state information and the provided external knowledge for joint reasoning. Additionally, an Evaluator supplies intrinsic rewards for both the planner and the actor. Experimental results conducted in the BabyAI environments demonstrate the effectiveness of this innovative approach.

**Strengths:**

- This paper studies an interesting topic of using natural language as external knowledge for RL agents to improve exploration in reward sparse tasks.

- The writing is clear and easy to follow.

- PAE showed strong performance in the experiments.

**Weaknesses:**

- The PAE model comprises three primary components, each with distinct hyper-parameters. Managing these varied parameters can be challenging in real applications.

- Since it depends on special environments and predefined knowledge template, PAE is not generalizable to a broader range of tasks.

**Questions:**

- What is the difference in wall-clock running time between PAE and the other baseline models?

- Would it be possible to extract latent knowledge from a collected dataset in a pre-training stage and replace the template-based knowledge in PAE?

---

> ### Author Response · Authors · 2023-11-18
> **Response to reviewer umQ7 (Part 1/2)**
>
> Thank you for your constructive comments and suggestions, which have greatly enhanced the quality of our manuscript. We have incorporated them into the revision. Below, we provide a point-by-point response to your comments.
>
> > Comment 1 :  The PAE model comprises three primary components, each with distinct hyper-parameters. Managing these varied parameters can be challenging in real applications.
>
> **Response:**
>
> We fully understand your concerns. As mentioned in Appendix A.4.1, our implementation of PAE and the reproduction of the baseline algorithm both utilize TorchBeast. Consequently, most hyperparameters align with those in TorchBeast's version, minimizing the need for extensive tuning. For new modules or structures unique to PAE, we conducted a grid search to determine their hyperparameters. The scope of this search and the final hyperparameters are detailed in Appendix A.3.4. Additionally, we have uploaded our source code in the appendix to facilitate easy reproduction of our work by other researchers.
>
> > Comment 2 :  Since it depends on special environments and predefined knowledge template, PAE is not generalizable to a broader range of tasks.
>
> **Response:**
>
> To address your concerns, we conducted additional experiments on MiniHack\[1\] to demonstrate that **PAE is scalable and does not rely on predefined knowledge templates**. MiniHack has a more complex state and action space and also possesses direct access to real natural language feedback ([Screenshot of Minihack Language](https://ibb.co/mDzz6Rm)) that human players receive during gameplay. We incorporate this systematic feedback into the PAE as external knowledge, making it more natural and realistic, rather than relying on specific templates. For additional information on the MiniHack task, please refer to Section 5 and Appendix A.2 of the revised version.
>
> We conducted comparisons between PAE and the main baselines across two task types, **totaling five environments within the MiniHack**. For your convenience, we have included a list of the results below and training curves ([Screenshot of Training-curves](https://imgbb.com/QJKqdxP)). For detailed information on the new MiniHack tasks and additional experimental results, please refer to Section 5 and Appendix A.2 in our revised version.
>
> Table 1:  Comparison of PAE and baseline methods in MiniHack tasks. (mean ± std, convergence steps)
>
> |           | LavaCross-Ring        | LavaCross-Potion      | LavaCross-Full        | Mutiroom-N4-Monster   | River-Monster         |
> |-----------|:-----------------------:|:-----------------------:|:-----------------------:|:-----------------------:|:-----------------------:|
> | Ours      | 1.00±0.001, 22M   | 0.99±0.002, 35M   | 1.00±0.001, 24M   | 0.72±0.020, >40M  | 0.13±0.018, >20M  |
> | L-AMIGo   | 0.57±0.223, >40M  | 0.58±0.214, >40M  | 0.45±0.111, >40M  | 0.31±0.035, >40M  | 0.09±0.016, >20M  |
> | AMIGo     | 0.46±0.116, >40M  | 0.54±0.048, >40M  | 0.44±0.101, >40M  | 0.29±0.026, >40M  | 0.08±0.009, >20M  |
> | ICM       | 0.00±0.000, >40M  | 0.00±0.000, >40M  | 0.00±0.000, >40M  | 0.18±0.199, >40M  | 0.12±0.028, >20M  |
> | IMPALA    | 0.00±0.000, >40M  | 0.00±0.000, >40M  | 0.00±0.000, >40M  | 0.00±0.000, >40M  | 0.00±0.000, >20M  |
> | RND       | 0.00±0.000, >40M  | 0.00±0.000, >40M  | 0.00±0.000, >40M  | 0.00±0.000, >40M  | 0.00±0.000, >20M  |
>
> Each entry in Table 1 consists of two numbers: the first denotes the average extrinsic reward achieved, and the second represents the minimum number of stabilizing steps required to achieve a reward plateau, where further significant increases are no longer observed. **The results demonstrate that our PAE approach exhibits good scalability and performs well without relying on predefined knowledge, i.e., without specific templates.**

---

> ### Author Response · Authors · 2023-11-18
> **Response to reviewer umQ7 (Part 2/2)**
>
> > Comment 3 :  What is the difference in wall-clock running time between PAE and the other baseline models?
>
> **Response:**
>
> Thank you very much for your suggestion. Wall-clock running time is indeed crucial for evaluating algorithm performance.
> To ensure a fair comparison and exclude interference, all algorithms must run independently on the same configured machines. Due to time constraints, we only report the wall-clock running time of different algorithms on KeyCorrS5R3 in Table 1.
>
> Table 2: Wall-clock running time for PAE and other baseline algorithms on KeyCorrS5R3
>
> |  Model  |  PAE (Ours)  |  L-AMIGo  |  AMIGo  |  ICM  |  IMPALA  |  RND  |
> | --- | --- | --- | --- | --- | --- | --- |
> |  Time (hours)  |  18.9   |  17.3   |  25.7   |  20.0   |  11.8   |  20.7   |
>
> As demonstrated in Table 2, our PAE method requires comparable training time to the baselines. This similarity is due to both our PAE and baseline algorithms being built on the TorchBeast implementations, with our method not introducing significant additional computation. Therefore, it does not significantly increase wall-clock running time.
>
> > Comment 4 :  Would it be possible to extract latent knowledge from a collected dataset in a pre-training stage and replace the template-based knowledge in PAE?
>
> **Response:**
>
> Yes, It would be. In fact, it aligns with our current approach, as mentioned in Section 4.1, where we utilize a pre-trained language model (i.e., BERT-base) to encode and generate knowledge embeddings. We use template-based knowledge solely because the BabyAI environment only provides this type of knowledge. In our further experiments with the new MiniHack environment, we incorporate real natural language feedback from human players during gameplay as external knowledge, eliminating the need for specific templates. The supplemental experiments show that our PAE approach remains superior. Detailed experimental results and non-template-specific knowledge from MiniHack are presented in Section 5 and Appendix A.2 of the revised version.

---

> ### Author Response · Authors · 2023-11-21
> **Have we addressed your concerns?**
>
> Thanks again for your time and effort in reviewing our paper! As the discussion period is coming to a close, we would like to know if we have resolved your concerns expressed in the original reviews. We remain open to any further feedback and are committed to making additional improvements if needed. If you find that these concerns have been resolved, we would be grateful if you would consider reflecting this in your rating of our paper : )

---

> > ### Comment · Reviewer_umQ7 · 2023-11-21
> >
> > Thanks for the reply. The responses mostly address my concerns and I will raise my rating to 6.

---

> > > ### Author Response · Authors · 2023-11-21
> > >
> > > Thank you very much for recognizing our work! We're pleased to hear that our clarifications have positively influenced your evaluation. We appreciate the time and effort you've invested in reviewing our manuscript and are grateful for your decision to raise your rating!
> > > --Best wishes from all the authors :)

---

### Official Review · Reviewer_rZvG · 2023-10-30

**Soundness:** 3 good
**Presentation:** 3 good
**Contribution:** 3 good
**Rating:** 6
**Confidence:** 4

**Summary:**

This paper introduces PAE: Planner-Actor-Evaluator, a novel framework for teaching agents to learn to absorb external knowledge. PAE integrates the Planner’s knowledge-state alignment mechanism, the Actor’s mutual information skill control, and the Evaluator’s adaptive intrinsic exploration reward. PAE aims to achieve cross-modal information fusion for the actor’s decision.

**Strengths:**

1.This work teaches agents to leverage external knowledge and approach optimal solutions faster in sparse reward environments.
2.This work introduces LLM ideas to RL problems, which is very interesting and novel.
3.It can be applied to the current actor-critic approaches.

**Weaknesses:**

1.As authors pointed, the current knowledge relies on the specific template, which lacks naturalness and does not fully use the huge ability of the LLM. It is an obvious weakness for this paper.
2.The current knowledge needs human labeling following the specific template, which is of labor consumption.
3.I suggest the definition and description of the alignment loss can be placed in the main text, which I finally find in the supplementary.

**Questions:**

1.The authors use the cross-attention as the alignment, how about the current alignment work like CLIP and ImageBind? Do the authors consider to use this alignment?
2.In Figure 4, I think there is unfair to compare with most of these baselines, since they do not have or use the language guidance. The L-AMIGo is the only one considering language. Are there any other similar work that the authors can compare? The current comparison is not very strong to highlight your work.

---

> ### Author Response · Authors · 2023-11-18
> **Response to reviewer rZvG (Part 1/3)**
>
> We appreciate your positive feedback and constructive comments, which have been greatly helpful in enhancing the quality of our manuscript. Below, we provide a point-by-point response to your comments.
>
> > Comment 1 :  As authors pointed, the current knowledge relies on the specific template, which lacks naturalness and does not fully use the huge ability of the LLM. It is an obvious weakness for this paper
>
> **Response:**
>
> We fully understand your concerns. We mainly focus on how external knowledge guides agent exploration and therefore directly adopt the pre-defined knowledge set provided by BabyAI without additional processing.
>
> To address your concerns, we have extended our PAE approach to the more challenging MiniHack task \[2\]. In addition to having a more complex state space and action space, MiniHack provides direct access to real natural language feedback ([Screenshot of Minihack Language](https://ibb.co/mDzz6Rm)) received by human players during gameplay. We integrate this feedback into PAE as external knowledge, **enhancing its naturalism and realism without the need for specific templates**. For more information on MiniHack tasks, see Section 5 and Appendix A.1 in the revised version.
>
> We conducted comparisons between PAE and the main baselines across two task types, **totaling five environments within the MiniHack**. For your convenience, we have included a list of the results below and training curves ([Screenshot of Training-curves](https://imgbb.com/QJKqdxP)). For detailed information on the new MiniHack tasks and additional experimental results, please refer to Section 5 and Appendix A.2 in our revised version.
>
> Table 1:  Comparison of PAE and baseline methods in MiniHack tasks. (mean ± std, convergence steps)
>
> |           | LavaCross-Ring        | LavaCross-Potion      | LavaCross-Full        | Mutiroom-N4-Monster   | River-Monster         |
> |-----------|:-----------------------:|:-----------------------:|:-----------------------:|:-----------------------:|:-----------------------:|
> | **Ours**      | **1.00±0.001, 22M**   | **0.99±0.002, 35M**   | **1.00±0.001, 24M**   | **0.72±0.020**, >40M  | **0.13±0.018**, >20M  |
> | L-AMIGo   | 0.57±0.223, >40M  | 0.58±0.214, >40M  | 0.45±0.111, >40M  | 0.31±0.035, >40M  | 0.09±0.016, >20M  |
> | AMIGo     | 0.46±0.116, >40M  | 0.54±0.048, >40M  | 0.44±0.101, >40M  | 0.29±0.026, >40M  | 0.08±0.009, >20M  |
> | ICM       | 0.00±0.000, >40M  | 0.00±0.000, >40M  | 0.00±0.000, >40M  | 0.18±0.199, >40M  | 0.12±0.028, >20M  |
> | IMPALA    | 0.00±0.000, >40M  | 0.00±0.000, >40M  | 0.00±0.000, >40M  | 0.00±0.000, >40M  | 0.00±0.000, >20M  |
> | RND       | 0.00±0.000, >40M  | 0.00±0.000, >40M  | 0.00±0.000, >40M  | 0.00±0.000, >40M  | 0.00±0.000, >20M  |
>
> Each entry in Table 1 consists of two numbers: the first denotes the average extrinsic reward achieved, and the second represents the minimum number of stabilizing steps required to achieve that reward. The results indicate that our PAE method performs superiorly without relying on predefined knowledge (i.e., no specific templates). The results of our complementary experiments have addressed the limitation regarding specific templates, and we have updated the discussion in the revised version.
>
> > Comment 2 :  The current knowledge needs human labeling following the specific template, which is of labor consumption.
>
> **Response:**
>
> We apologize for the confusion caused by our unclear presentation. Currently, knowledge in BabyAI is automatically provided by the environment and doesn't require additional human labeling. To address your concerns further, we've added the MiniHack task, which provides real natural language feedback ([Screenshot of Minihack Language](https://ibb.co/mDzz6Rm)) received by the agent during gameplay, and **no specific template requires human labeling**. Extended experiments in MiniHack demonstrate that our PAE can handle non-template-specific knowledge in natural language form. You can find the list of knowledge in MiniHack and the experiment results in Appendix A.2 of the revised version.
>
> > Comment 3 :  I suggest the definition and description of the alignment loss can be placed in the main text, which I finally find in the supplementary.
>
> **Response:**
>
> Thank you very much for your valuable suggestion. Following your advice, we have relocated the definition and description of the alignment loss to the main text in our revised manuscript. This adjustment ensures that this crucial information is more readily accessible and appropriately highlighted.

---

> ### Author Response · Authors · 2023-11-18
> **Response to reviewer rZvG (Part 2/3)**
>
> > Comment 4 :  The authors use the cross-attention as the alignment, how about the current alignment work like CLIP and ImageBind? Do the authors consider to use this alignment?
>
> **Response:**
>
> This is an interesting and worth exploring question. If we go a little deeper into the details, we can **unify** the mechanism of alignment in PAE, CLIP, and ImageBind. In PAE, we employ Scaled-CrossAttention for aligning knowledge $\textbf{k}$ and state $\textbf{s}$ , generating the Attention Map between the Query $Q=W\_Q\cdot \textbf{k}$ and the Key $K=W\_K\cdot \textbf{s}$ . CLIP and ImageBind achieve cross-modal ( $\textbf{m}_1$ and  $\textbf{m}_2$) alignment by utilizing a similar Key-Query manner, generating an Relation Map between the Query $Q=W\_Q\cdot \textbf{m}_1$ and the Key $K=W\_K\cdot \textbf{m}_2$ .
>
>  The primary distinction among these three methods lies in their use of different normalization techniques.
>
>  $\text{PAE}: \quad  \quad \quad \text{Attention} = \text{SoftMax}(Q\cdot K^T/ \sqrt{d\_k})$
>
>  $\text{CLIP}: \quad \quad \quad \text{Relation} = \text{SoftMax}(\text{norm}(Q)\cdot \text{norm}(K^T)) \quad  $
>
>  $\text{ImageBind}: \quad \text{Relation} = \text{SoftMax}(Q\cdot K^T/\tau)  $
>
> Specifically, PAE normalizes using the dimension of the Key vector, while CLIP normalizes both Key and Query, and ImageBind employs temperature coefficients $\tau$ for normalization.
>
> CLIP and ImageBind may offer a significant advantage as they are pre-trained on large natural image datasets. However, there is a domain gap between natural images and unrealistic virtual game graphics. More importantly, our environments (BabyAI and MiniHack) and various baselines all use vectorized inputs, so CLIP and ImageBind are not suitable for direct porting to our task, which is why we chose the learnable Scaled-CrossAttention alignment mechanism. Thanks to your insights, we have included a discussion on these two approaches in Appendix A.1 of revised version.

---

> ### Author Response · Authors · 2023-11-18
> **Response to reviewer rZvG (Part 3/3)**
>
> > Comment 5 :  In Figure 4, I think there is unfair to compare with most of these baselines, since they do not have or use the language guidance. The L-AMIGo is the only one considering language. Are there any other similar work that the authors can compare? The current comparison is not very strong to highlight your work.
>
> **Response:**
>
> Thank you very much for your valuable suggestion. We have followed your suggestion and introduced another recent Language-Instructed Algorithm called L-NovelD \[3\] for comparison with our PAE method. L-NovelD combines natural language with an intrinsic motivation approach to reward states described in natural language that transitions from low to high novelty. Mu et al. \[3\]  demonstrated that L-NovelD outperforms several popular intrinsic motivation methods, including NovelD, RND, and others, in the BabyAI tasks. Detailed information about the L-NovelD method can be found in Section 5 and Appendix A.4.2 of the revised version.
>
> We conducted a comparative analysis of the PAE and L-NovelD methods across the six BabyAI environments. The table below and training curves ([Screenshot of Training-curves](https://imgbb.com/QJKqdxP)) present results for the added L-NovelD and PAE comparisons. For a comprehensive view of experimental results and training curves, please refer to Section 5 of the revised version.
>
> Table 2: Comparison of PAE and baseline methods (mean ± std, convergence steps)
>
> |  Model  |  KEYCORRS3R3  |  KEYCORR43R3  |  KEYCORRS5R3  |  OBSTRMAZE1D1  |  OBSTRMAZE2D1HB  |  OBSTRMAZE1Q  |
> | --- | --- | --- | --- | --- | --- | --- |
> |  **Ours**  |  **0.89**$\pm$**0.002**, **6M**  |  **0.92**$\pm$**0.005**, **30M**  |  **0.94**$\pm$**0.001**, **90M**  |  **0.93**$\pm$**0.004**, **6M**  |  **0.87**$\pm$**0.018**, **150M**  |  **0.89**$\pm$**0.006**, **150M**  |
> |  L-AMIGo  |  0.86 $\pm$ 0.040, 12M  |  0.90 $\pm$ 0.011, 60M  |  0.93 $\pm$ 0.016, 160M  |  0.90 $\pm$ 0.030, 20M  |  0.23  $\pm$0.403, \>200M  |  0.47 $\pm$ 0.029, \>300M  |
> |  L-NovelD  |  0.63  $\pm$0.097, \>20M  |  0.15 $\pm$ 0.195, \>60M  |  0.55 $\pm$ 0.484, \>200M  |  0.79 $\pm$ 0.255, \>20M  |  0.57 $\pm$ 0.164, \>200M  |  0.60 $\pm$ 0.107, \>200M  |
> |  AMIGo  |  0.43 $\pm$ 0.246, \>20M  |  0.22 $\pm$ 0.343, \>60M  |  0.43  $\pm$0.599, \>200M  |  0.45  $\pm$0.366, \>20M  |  0.10 $\pm$ 0.175, \>200M  |  0.18 $\pm$ 0.023, \>300M  |
> |  ICM (Curiousity)  |  0.04 $\pm$ 0.052, \>20M  |  0.30 $\pm$ 0.511, \>60M  |  0.00  $\pm$0.000, \>200M  |  0.31  $\pm$0.542, 15M  |  0.00  $\pm$0.000, \>200M  |  0.00 $\pm$ 0.000, \>300M  |
> |  IMPALA  |  0.00 $\pm$ 0.000, \>20M  |  0.00 $\pm$0.000, \>60M  |  0.00 $\pm$ 0.000, \>200M  |  0.00 $\pm$ 0.006, \>20M  |  0.00 $\pm$ 0.000, \>200M  |  0.00 $\pm$ 0.000, \>300M  |
> |  RND  |  0.00 $\pm$ 0.000, \>20M  |  0.00 $\pm$ 0.000, \>60M  |  0.00 $\pm$ 0.000, \>200M  |  0.00 $\pm$ 0.000, \>20M  |  0.00 $\pm$ 0.000, \>200M  |  0.00 $\pm$ 0.000, \>300M  |
>
> \*Due to very limited time, we have so far completed the adaptation and training of L-NovelD on 6 environments in BabyAI, and the training in the 5 additional MiniHack environments is still running. We will update our full experiments in the camera ready version.
>
> Each entry in Table 1 consists of two numbers: the first denotes the average extrinsic reward achieved, and the second represents the minimum number of stabilizing steps required to achieve a reward plateau, where further significant increases are no longer observed. The results demonstrate that our PAE method still outperforms the L-NovelD method, which also utilizes natural language as an aid.
>
> **Ref :**
>
> \[1\] Alec Radford, et al. "Learning transferable visual models from natural language supervision." ICML2021.
>
> \[2\] Rohit Girdhar, et al. "Imagebind: One embedding space to bind them all." CVPR2023.
>
> \[3\] Jesse Mu, et al. "Improving Intrinsic Exploration with Language Abstractions." NeurIPS 2022.

---

> ### Author Response · Authors · 2023-11-21
> **Have we addressed your concerns?**
>
> Thanks again for your time and effort in reviewing our paper! As the discussion period is coming to a close, we would like to know if we have resolved your concerns expressed in the original reviews. We remain open to any further feedback and are committed to making additional improvements if needed. If you find that these concerns have been resolved, we would be grateful if you would consider reflecting this in your rating of our paper :)

---

> > ### Comment · Reviewer_rZvG · 2023-11-22
> > **Reply to the authors**
> >
> > Thanks for you effort. The authors provide detailed results and thinking. I will remain my score. That is a relatively good paper.

---

> > > ### Author Response · Authors · 2023-11-22
> > >
> > > Thank you very much for your support! We appreciate the time and effort you've invested in reviewing our manuscript, and your comments have helped us improve our paper!
> > > --Best wishes from all the authors :)

---

### Official Review · Reviewer_2pSJ · 2023-10-31

**Soundness:** 3 good
**Presentation:** 3 good
**Contribution:** 2 fair
**Rating:** 6
**Confidence:** 4

**Summary:**

The paper presents PAE (Planner-Actor-Evaluator), a novel framework for incorporating external knowledge in reinforcement learning (RL), with the aim of enabling efficient exploration and skill acquisition in sparse reward environments. The framework is built upon three main components: the Planner, the Actor, and the Evaluator. The Planner aligns knowledge in natural language with states in the environment, the Actor integrates external knowledge with internal strategies, and the Evaluator computes intrinsic rewards to guide independent updates of the Planner and Actor. Experiments on six BabyAI environments demonstrated that PAE significantly outperforms existing methods in terms of exploration efficiency and interpretability.

**Strengths:**

- Originality: The paper proposes a new framework addressing the issue of knowledge acquisition, integration, and updating in RL. PAE stands out with its ability to use natural language as a knowledge source and to progressively adapt the difficulty of acquired knowledge.
- Quality: The proposed PAE framework is designed with clear objectives, tackling key challenges that arise when training agents to absorb external knowledge and improve their capabilities.
- Clarity & Significance: The experiments conducted in multiple challenging environments provide strong evidence of the framework's effectiveness, its generalization across tasks, and superiority compared to existing methods. The paper is well-written and provides meaningful insights into the relationship between knowledge and environment states, making PAE a valuable contribution to the RL community.

**Weaknesses:**

I believe that the main limitation of this paper is that the PAE algorithm relies on a pre-defined knowledge set, although the authors have proactively mentioned this in the limitations section. There are already quite a few LLM-based agents that do not require pre-defined external knowledge and are able to complete various tasks well by relying solely on the LLM's inherent commonsense knowledge [1] or by actively gathering, summarizing, or reflecting on dynamic knowledge throughout the task [2]. However, the experimental section does not compare these methods. Although the authors mention that these methods may have higher training/inference costs, I think that given the current popularity of LLM-based agents, this comparison is essential, and higher training/inference costs are not a compelling reason to avoid it. Otherwise, it is difficult to justify the necessity of pre-defined external knowledge and the rationality of the basic settings of this paper.

---
[1] Carta, Thomas, et al. "Grounding Large Language Models in Interactive Environments with Online Reinforcement Learning." ICML 2023.

[2] Chen, Liting, et al. "Introspective Tips: Large Language Model for In-Context Decision Making." arXiv preprint arXiv:2305.11598 (2023).

**Questions:**

The main questions have been mentioned in the `Weaknesses` section.

---

> ### Author Response · Authors · 2023-11-18
> **Response to reviewer 2pSJ (Part 1/2)**
>
> Thank you for your encouraging words and constructive feedback. We appreciate your time spent reviewing our paper and have provided point-by-point responses to your comments below.
>
> > Comment 1 :  I believe that the main limitation of this paper is that the PAE algorithm relies on a pre-defined knowledge set, although the authors have proactively mentioned this in the limitations section.
>
> **Response:**
>
> We fully understand your concerns. We mainly focus on how external knowledge guides agent exploration and therefore directly adopt the pre-defined knowledge set provided by BabyAI without additional processing.
>
> To address your concerns, **we have applied our PAE approach to the more challenging MiniHack task**\[1\] ([Screenshot of Minihack](https://ibb.co/sK1n2Fb)). In addition to having a more complex state space and action space, MiniHack provides direct access to real natural language feedback ([Screenshot of Minihack Language](https://ibb.co/mDzz6Rm)) received by human players during gameplay. We incorporate this system feedback into PAE as external knowledge, which is **more natural and realistic, without relying on specific templates**. For further details on the MiniHack tasks, please refer to Section 5 and Appendix A.2 of the revised version.
>
> We conducted comparisons between PAE and the main baselines across two task types, **totaling five environments within the MiniHack**. For your convenience, we have included a list of the results below  and training curves ([Screenshot of Training-curves](https://imgbb.com/QJKqdxP)). For detailed information on the new MiniHack tasks and additional experimental results, please refer to Section 5 and Appendix A.2 in our revised version.
>
> Table 1:  Comparison of PAE and baseline methods in MiniHack tasks. (mean ± std, convergence steps)
>
> |           | LavaCross-Ring        | LavaCross-Potion      | LavaCross-Full        | Mutiroom-N4-Monster   | River-Monster         |
> |-----------|:-----------------------:|:-----------------------:|:-----------------------:|:-----------------------:|:-----------------------:|
> | **Ours**      | **1.00±0.001, 22M**   | **0.99±0.002, 35M**   | **1.00±0.001, 24M**   | **0.72±0.020**, >40M  | **0.13±0.018**, >20M  |
> | L-AMIGo   | 0.57±0.223, >40M  | 0.58±0.214, >40M  | 0.45±0.111, >40M  | 0.31±0.035, >40M  | 0.09±0.016, >20M  |
> | AMIGo     | 0.46±0.116, >40M  | 0.54±0.048, >40M  | 0.44±0.101, >40M  | 0.29±0.026, >40M  | 0.08±0.009, >20M  |
> | ICM       | 0.00±0.000, >40M  | 0.00±0.000, >40M  | 0.00±0.000, >40M  | 0.18±0.199, >40M  | 0.12±0.028, >20M  |
> | IMPALA    | 0.00±0.000, >40M  | 0.00±0.000, >40M  | 0.00±0.000, >40M  | 0.00±0.000, >40M  | 0.00±0.000, >20M  |
> | RND       | 0.00±0.000, >40M  | 0.00±0.000, >40M  | 0.00±0.000, >40M  | 0.00±0.000, >40M  | 0.00±0.000, >20M  |
>
> Each entry in Table 1 consists of two numbers: the first denotes the average extrinsic reward achieved, and the second represents the minimum number of stabilizing steps required to achieve a reward plateau, where further significant increases are no longer observed. The results indicate that our PAE method performs superiorly without relying on predefined knowledge (i.e., no specific templates). The results of our complementary experiments have addressed the limitation regarding specific templates, and we have updated the discussion in the Appendix A.1.

---

> ### Author Response · Authors · 2023-11-18
> **Response to reviewer 2pSJ (Part 2/2)**
>
> > Comment 2 :  Although the authors mention that these methods may have higher training/inference costs, I think that given the current popularity of LLM-based agents, this comparison is essential, and higher training/inference costs are not a compelling reason to avoid it. Otherwise, it is difficult to justify the necessity of pre-defined external knowledge and the rationality of the basic settings of this paper.
>
> **Response:**  This is an interesting and worth exploring question. As you mentioned, LLM-based agents are gaining popularity in solving RL tasks, and very recent studies have shown considerable potential. Thank you for the two related works \[1,2\] that provided us with some new insights. As far as we know, there are currently two main categories of LLM-based agents for decision-making: 1. Fine-tuning LLMs using RL for decision-making, and 2. Employing LLMs directly for decision-making as plug-ins. These differ significantly from our approach, which **focuses on using external knowledge to enhance RL algorithms**. In contrast, **recent LLM-based agent studies primarily showcase the abilities of LLMs or expand their capabilities using RL methods**. In our research, LLMs are a tool to aid agents in understanding the semantics of knowledge.
>
> We appreciate the new insights you've provided and have tried our best to compare the methods you mentioned with PAE. However, we found that these LLM methods **currently only accept textual state inputs**, which means that they can currently only be used in customized environments. This limitation is clearly stated in the Abstraction of both articles: 'Using an interactive textual environment designed to study...', 'Experiments involving over 100 games in TextWorld...'.
>
> On the other hand, we spent some time trying to run the open-source code in \[1\] ([https://github.com/flowersteam/Grounding\_LLMs\_with\_online\_RL](https://github.com/flowersteam/Grounding_LLMs_with_online_RL)) by converting the environment we used in BabyAI to textual input. Our new experimental results show that this approach **does not converge in the BabyAI Key Corridor task** we used (using the open source code, all configurations are kept the same). By carefully reading the original paper, we found that Appendix B.1 (page 20 ([Screenshot](https://ibb.co/z5n3v6K)) in the original paper \[1\] mentions that "GFlan-T5 has not found yet any robust strategy for the OpenDoor task (being the hardest as the agent must find the right key and discover that the action "toggle" opens the door) in the relatively short allocated time." This is also consistent with our reproduction results.
>
> While for our task difficulty settings, "OpenDoor" is only one of many subtasks that the agent must complete in the correct order to receive the final reward (see Figures 5, 10, and 11 in our paper), and our PAE approach accomplishes these tasks well. We believe that the capabilities of large language models in unknown exploration tasks **might be overestimated**, but this remains open for community discussion. Regardless, **this question is very inspiring, thought-provoking and discussion-worthy**, and we will delve into the strengths and limitations of the LLM-Agent in the discussion section. Thanks for your valuable contribution to the depth of our work.
>
> **Summary**: After experimental validation and thorough review of related work, we conclude that: 1. The current focus of LLM-based agent approaches differs from our PAE method; **the former aims to enhance LLM agent capabilities, while the latter focuses on improving RL agent capabilities**. 2. The LLM agent approach does not surpass our PAE method in certain tasks (as evidenced by our BabyAI task tests) and is limited to text input only. We have incorporated these insights , as well as related discussions, into Appendix A.1 of the revised version.
>
> **Ref :**
>
> \[1\] Carta, Thomas, et al. "Grounding Large Language Models in Interactive Environments with Online Reinforcement Learning." In ICML 2023.
>
> \[2\] Chen, Liting, et al. "Introspective Tips: Large Language Model for In-Context Decision Making." arXiv preprint arXiv:2305.11598, 2023.

---

> > ### Comment · Reviewer_2pSJ · 2023-11-20
> >
> > Thank you very much for the author's response, as it addressed my main concerns. The MiniHack experiment and its results provided addressed what I believe to be the most significant limitation of the PAE method in the use of external knowledge. You have also made a thorough comparison with the two baselines [1,2] I mentioned, and drew a preliminary conclusion that "the capabilities of large language models in unknown exploration tasks might be overestimated." I think this is quite an important finding for language/AI/LLM agents, and I'm very grateful for the your additional experiment.
> >
> > However, I'm somewhat puzzled as to why the messages in Table 4 help facilitate exploration. Unlike the knowledge shown in BabyAI tasks in Table 3, the messages in Table 4 do not contain explicit instructions to guide the agent in exploration/decision-making and are mostly feedback on interactions or descriptions of the environment. I hope you could provide some intuitive explanations, and it would be great if they could be combined with intermediate results from the MiniHack tasks. Thank you!

---

> ### Author Response · Authors · 2023-11-20
> **New visualization results + intuitive explanation**
>
> Thank you for your prompt response and also thank you for recognizing our supplementary experiment, we are glad that we have addressed your main concern.
>
> Your puzzle about the MiniHack's messages is actually a good question, a corresponding intuitive discussion may better clarify our key ideas to the readers. So, let's make a detailed explanation for it! (Besides, we also provide some **intermediate experiment results** to make our explanations more intuitive)
>
> **(1) The knowledge presented in Table 3 for BabyAI and in Table 4 for MiniHack can be regarded as target states**. For example, in MiniGrid, the message `"go to the door"` can be interpreted as `(in this state), go to the door`. In MiniHack, the message `"the o is killed!"` can be interpreted as `(in this state), kill the o!`. We can view these system text descriptions as the **target states** that the Planner **requires the Actor to achieve** after taking a series of actions. Consequently, the Actor must either explore to try new actions to achieve these goals or extract useful strategies from the past actions (trajectories) towards these goals.
>
> **(2) Much of the knowledge in MiniHack is not directly contribute to task completion.** Given that MiniHack's knowledge is derived from actual player feedback during gameplay, it naturally includes some irrelevant or emotionally messages, such as 'ouch!' and 'never mind'. A rough estimate is provided here: an agent capable of solving the Lavacross task will encounter approximately 80 messages, of which only 6-10 (8-13%) are necessary for a successful trajectory. **This requires the Planner to semantically understand these environment messages, filter and exclude those useless knowledges**.
>
> **We also provide some intermediate experiment results for better understanding:** ([screenshot of intermedia experiment](https://ibb.co/WF1GMX4)). We use "LavaCross-Ring" in MiniHack as an example.  Due to the complexity of the MiniHack environment, the Planner adopts a more exploratory curriculum. Initially, it presents exploration-based knowledge, such as `you see a uncursed flint stone`, to help the Actor get familiar with the environment. At a certain stage, the Planner observes the presence of a ring in the environment `you see here a ring`, dedicates more effort to manipulating the ring `what do you want to put on?`, and eventually asks the Actor to reach a state of deciding how to place the ring on the correct finger.  `which ring finger, right or left?` to accomplish the task of escaping the dungeon `descend`. **The Planner's curriculum is summarized as follows: "Explore the environment" -> "Find the ring" -> "Put on the ring" -> "Put on the ring on the ringht finger" -> "Descend (success)".**
>
> Consequently, we have included the above additional explanations and discussions in Appendix A.2 to clarify how the knowledge works. We hope our response addresses your confusion.
>
> We appreciate the new insights you've provided and your valuable contribution to the depth of our work! Please let us know if you have any further questions.
>
> If you find that these concerns have been resolved, we would be grateful if you would consider reflecting this in your rating of our paper : )

---

> > ### Comment · Reviewer_2pSJ · 2023-11-21
> >
> > Thank you very much for your explanation and additional experiments. Treating knowledge as a goal is indeed an intuitive and interesting interpretation. Considering that your response addressed most of my concerns, I will raise the review score to 6.
> >
> > Additionally, I have a minor question. The knowledge base for BabyAI seems to be pre-existing, but MiniHack appears to require constant collection during the training process. How do you ensure that the algorithm can collect enough messages? It seems like this still requires encouraging agents to explore, but enhancing exploration capabilities is also a goal that PAE aims to achieve in the MiniHack tasks. There appears to be some contradiction between these two aspects.

---

> ### Author Response · Authors · 2023-11-21
> **Following discussion + Best Wishes from the authors**
>
> Thank you very much for recognizing our work!
>
> Regarding your following minor question, a possible intuitive explanation is that: in the early stages of training, the Planner actually does not know what kind of goal would lead the Actor to the ultimate reward. Therefore, it might randomly try different messages collected online previously. For example, `you see an uncursed flint stone` is actually not necessary for completing the dungeon level in "LavaCross-Ring". However, the Planner finds that these messages help enhance the Actor's exploration from its past proposed trajectories. Thus, it reinforces these goals in the early stages. But as the curriculum progresses and the Actor's abilities improve, the Planner will gradually focus on how to lead the Actor to the completion of the main task.
>
> It's noteworthy that whether it's in the early stages when the Actor takes random-like actions, or later when the Actor is trying to complete the Planner's curriculum, there's a high probability that the Actor may trigger new messages. These messages will be instantly collected by the Planner, and these new messages may be organized for the future curriculum. This forms **a positive cycle of learning, exploration, and curriculum customization.**
>
> We believe that the question of how to enable agents to gather both diverse and informative knowledge during training is quite an open-discussion topic, and we will conduct more in-depth research on this in the near future. We appreciate the inspiring suggestions you have provided.
>
> Meanwhile, we want to thank you for your thoughtful review and insights into our work, which have significantly improved the quality of our paper. Your time and effort are greatly appreciated!
>
> ---Respect and Best Wishes from the authors

---

> > ### Comment · Reviewer_2pSJ · 2023-11-21
> >
> > Thank you for your comprehensive and insightful response to my previous concern. I appreciate the additional intuitive explanation you provided, which further clarifies the dynamics between the `Planner` and the `Actor` in the early stages of training. I also find the concept of a positive cycle of learning, exploration, and curriculum customization intriguing and a valuable aspect of your approach. I have raised my score to 6.

---

### Official Review · Reviewer_qGp8 · 2023-10-31

**Soundness:** 4 excellent
**Presentation:** 3 good
**Contribution:** 3 good
**Rating:** 8
**Confidence:** 4

**Summary:**

They propose a novel framework for sparse-reward, long-horizon RL tasks consisting of Planner, Actor and Evaluator. Planner provides guidance/knowledge to the Actor in natural language. Actor is incentivized to explore states aligned with that guidance. Evaluator evaluates if Actor is able to accomplish a sub-task given Planner’s guidance and provides intrinsic rewards to both Planner and Actor. The Actor is given a +1 on completing a sub-task and the Planner is given a reward of +alpha if the Actor completed the sub-task under an adaptive threshold number of steps. Otherwise, Planner gets a -beta. The core idea is that the Planner must generate increasingly harder knowledge/tasks for the Actor based on Actor’s current abilities. They use BabyAI for experiments. Both Planner and Actor also receive external environmental rewards.

**Strengths:**

1.	This paper is attempting to bring natural language instruction to curriculum-driven RL by adding in Planner to a modified version of Actor Critic
2.	Their encoders use cross-attention to align instruction knowledge with current environment state embedding to encourage selection of the most relevant knowledge and exploration of relevant states by actor. The actor is also rewarded based on the mutual information between knowledge and states.
3.	The paper is well structured even though the methodology is complex.
4.	The reward structure is intuitive to understand and clearly defined for each component.
5.	They compare against IMPALA, RND, ICM, Amigo, L-Amigo and have consistent improvement as well as good training stability across 6 BabyAI tasks
6.	The ablation studies are thoroughly done and establish the value added by the Planner.
7.	The Planner generated curriculum serves as a window into agent behavior

**Weaknesses:**

1.	BabyAI action space is quite limited (6 actions) and simplistic, so are the tasks since they are 2D grid world tasks.
2.	As pointed out by authors, current experiments have a knowledge oracle and a very specific template for knowledge peices. Also, planners can observe the whole world, which may be unrealistic for many real-world set up.
3.	Current experiments require well-defined sub-tasks as well as detection of sub-task completion. Baby AI tasks are about navigating a grid in a particular way and due to task and world structure, it is easy to detect sub-task completion.
4.	This method would need sub-task and knowledge alignment so as to detect if a particular piece of knowledge lead to completion of a particular sub-task. I don’t see how it can work any other way. This is a very hard thing to obtain.
5.	The evaluator currently is highly customized to BabyAI tasks.

**Questions:**

1. I would like to hear authors thoughts on weaknesses 3, 4 and 5 to be sure of my understanding of the paper's contributions
2. How did the authors reach the design for evaluator? What challenges if any, do the authors foresee in making the evaluator more general-purpose?

---

> ### Author Response · Authors · 2023-11-18
> **Response to reviewer qGp8 (Part 1/3)**
>
> Thank you very much for your constructive comments and suggestions. We have revised our paper accordingly. Below, we will provide detailed responses to each point.
>
> > Comment 1 : BabyAI action space is quite limited (6 actions) and simplistic, so are the tasks since they are 2D grid world tasks.
>
> **Response :**
> To address your concerns, we have extended our PAE approach to the **more challenging** MiniHack tasks \[1\]. MiniHack consists of procedurally generated tasks within a roguelike game, offering **a richer observation space** and up to **75 dimensions of structured and context-sensitive action space**, presenting huge challenges. The high complexity of this state-action space makes MiniHack **one of the most challenging sparse reward environments** to explore \[2\]. We conducted comparisons between PAE and the main baselines across two task types,**totaling five** **environments** (**LavaCross-Ring, LavaCross-Potion, LavaCross-Full, MultiRoom-N4-Monster, River-Monster**) within the MiniHack. For ease of comparison, we have included a list of the results below and training curves ([Screenshot of Training-curves](https://imgbb.com/QJKqdxP)). For detailed information on the new MiniHack tasks and additional experimental results, please refer to Section 5 and Appendix A.2 in our revised version.
>
> We have also provided ample experimental results to demonstrate the effectiveness and scalability of our work. Particularly, we have shown that our approach can be readily extended to more challenging environments.
>
> Table 1: Comparison of PAE and baseline methods in MiniHack tasks. (mean ± std, convergence steps)
>
> | | LavaCross-Ring | LavaCross-Potion | LavaCross-Full | Mutiroom-N4-Monster | River-Monster |
> |-----------|:-----------------------:|:-----------------------:|:-----------------------:|:-----------------------:|:-----------------------:|
> | **Ours** | **1.00±0.001, 22M** | **0.99±0.002, 35M** | **1.00±0.001, 24M** | **0.72±0.020**, >40M | **0.13±0.018**, >20M |
> | L-AMIGo | 0.57±0.223, >40M | 0.58±0.214, >40M | 0.45±0.111, >40M | 0.31±0.035, >40M | 0.09±0.016, >20M |
> | AMIGo | 0.46±0.116, >40M | 0.54±0.048, >40M | 0.44±0.101, >40M | 0.29±0.026, >40M | 0.08±0.009, >20M |
> | ICM | 0.00±0.000, >40M | 0.00±0.000, >40M | 0.00±0.000, >40M | 0.18±0.199, >40M | 0.12±0.028, >20M |
> | IMPALA | 0.00±0.000, >40M | 0.00±0.000, >40M | 0.00±0.000, >40M | 0.00±0.000, >40M | 0.00±0.000, >20M |
> | RND | 0.00±0.000, >40M | 0.00±0.000, >40M | 0.00±0.000, >40M | 0.00±0.000, >40M | 0.00±0.000, >20M |
>
> Each entry in Table 1 consists of two numbers: the first denotes the average extrinsic reward achieved, and the second represents the minimum number of stabilizing steps required to achieve a reward plateau, where further significant increases are no longer observed. The results indicate that our PAE method performs superiorly and is easy to be adapted to more challenging environments.
>
> > Comment 2.1: As pointed out by authors, current experiments have a knowledge oracle and a very specific template for knowledge peices.
>
> **Response :**
>
> Thank you very much for your comments. The second reason we conducted experiments on the MiniHack task is that, MiniHack provides **natural language knowledge** ([Screenshot of Minihack Language](https://ibb.co/mDzz6Rm)) which is real-world feedback received by human players or agents during gameplay, rather than specific knowledge template in BabyAI. Extended experiments in MiniHack ([Screenshot of Training-curves](https://imgbb.com/QJKqdxP)) demonstrate our PAE's ability to handle non-template-specific knowledge in natural language form and. its effectiveness, versatility, and scalability. A list of the knowledge from MiniHack and the experiment results can be found in Section 5 and Table 4 in the revised version.
>
> We mention these limitations in the manuscript because our research originally focused on how external knowledge guides agent exploration. So, it **directly followed the templates provided by BabyAI to express knowledge**. The results of our latest complementary experiments have addressed the limitation regarding specific templates, and we have updated more discussions in the revised version accordingly.
>
> Besides, we are very grateful for your suggestions to our work, which have significantly contributed to our exploration of the boundaries of the PAE's capabilities. Thank you very much.
>
> > Comment 2.2: Planners can observe the whole world, which may be unrealistic for many real-world set up.
>
> This is actually the default setting of the baseline methods in the BabyAI environment, and we strictly followed this setting for a fair comparison. To address your concern, in our newly introduced MiniHack task, both **the Planner and Actor utilize identical partial observation inputs.** This uniform setting is also maintained across all other baselines. Experimental results confirm that our PAE method can still consistently outperform other baselines.

---

> ### Author Response · Authors · 2023-11-18
> **Response to reviewer qGp8 (Part 2/3)**
>
> > Comment 3 :  Current experiments require well-defined sub-tasks as well as detection of sub-task completion. BabyAI tasks are about navigating a grid in a particular way and due to task and world structure, it is easy to detect sub-task completion.
>
> **Response:**
>
> Thank you very much for your comments. You are correct in understanding that the current experiment involves explicit sub-tasks and the detection of sub-task completion.  In fact, most of the subtasks provided in the BabyAI environment do not contribute to the completion of the final task, which, on average, only requires 6 to 12 messages (1-2% of all 652 messages) to complete \[3\]. The key to earning the final reward lies in **identifying useful subtasks and establishing the sequential relationships between them**.
>
> In Figure 5 of our paper, we visually demonstrate how PAE progressively learns this logical relationship, and Figure 10 illustrates the process that agents across various tasks must follow to achieve the final reward. Despite intrinsic motivation methods like AMIGo, L-AMIGo, etc., **also receiving cues about sub-tasks and sub-task completion detection, our PAE methods outperform them**, as indicated in Figure 4.
>
> Additionally, we claim that sub-tasks, like certain actions triggering specific text messages, are more and more common among games, namely "in-game guidance messages". These text messages, which already embedded in the status information, may not be fully utilized by previous baseline methods, limiting their performance. While in our work PAE, the Planner's role is to automatically align and organize these sub-tasks into its curriculum, aiding the Actor in better assimilating external knowledge and exploring the environment.
>
> Last but not least, our new MiniHack environment includes **Skill Acquisition Tasks, distinct from the navigation tasks in BabyAI** with less structured text knowledge, yet our PAE excels in grasping the sequential relationship between subtasks to complete the final task, surpassing other baselines. backing its effectiveness.
>
> > Comment 4 :  This method would need sub-task and knowledge alignment so as to detect if a particular piece of knowledge lead to completion of a particular sub-task. I don’t see how it can work any other way. This is a very hard thing to obtain.
>
> **Response:**
>
> We fully understand your concerns and apologize for any confusion arising from the lack of clear explanations in our paper. External knowledge indeed contributes to subtask completion detection, but this is not the full and crucial role played by external knowledge. Despite intrinsic motivation methods like AMIGo, L-AMIGo, etc., also receiving cues about sub-tasks and sub-task completion detection, our PAE methods outperform them, as indicated in Figure 4. The core innovation of our paper is how to utilize external knowledge to enhance the agent's exploration capabilities. We introduce a generalized framework, PAE: the **Planner** utilizes external knowledge for creating a progressive curriculum through feedback and guidance; the **Actor** aligns its current capabilities with the environment state with external knowledge's assistance, and the **Evaluator** employs external knowledge to detect subtask completion.
>
> > Comment 5 :  The evaluator currently is highly customized to BabyAI tasks. How did the authors reach the design for evaluator? What challenges if any, do the authors foresee in making the evaluator more general-purpose?
>
> **Response:** Thank you very much for this insightful question. The Evaluator's primary insight lies in providing guidance to both the Planner and the Actor through the design of intrinsic rewards. Compared to previous approaches, the Evaluator in PAE introduces two key innovations. Firstly, it leverages external knowledge to detect subtask completion, offering a more natural and comprehensible approach. Secondly, the Evaluator guides both the Planner and the Actor by encouraging the Planner to provide incremental guidance that is neither too easy nor too challenging, while motivating the Actor to complete tasks promptly under the Planner's guidance. **This design concept has broad applicability, and the Evaluator's key parameters are adaptive**, with a step threshold (t\*) that automatically grows as the agent's exploration ability improves, capped at 30% of the environment's maximum step count. To validate the generality of the Evaluator, we tested it in the MiniHack environment, where our PAE approach also exhibited superior performance ([Screenshot of Training-curves](https://imgbb.com/QJKqdxP), [Screenshot of Addtional-Results](https://ibb.co/Wnsf0Qw)). Detailed experimental results are presented in Section 5 and Appendix A.2 in our revised version.

---

> ### Author Response · Authors · 2023-11-18
> **Response to reviewer qGp8 (Part 3/3)**
>
> **Summary of our responses related to your comments:**
>
> *   We demonstrated on the more difficult MiniHack task that PAE can handle more complex action spaces.
>
> *   We demonstrated through supplementary experiments that the PAE method does not rely on specific templates.
>
> *   We have clarified and explained the role of external knowledge in PAE in the revised version.
>
> *   We clarify the Evaluator's design motivation and demonstrate its strong generalization through supplementary experiments.
>
>
> **Ref :**
>
> \[1\] Mikayel Samvelyan, et al. "MiniHack the Planet: A Sandbox for Open-Ended Reinforcement Learning  Research." In NeurIPS 2021.
>
> \[2\] Mikael Henaff, et al. "A Study of Global and Episodic Bonuses for Exploration in Contextual MDPs." In ICML 2023.
>
> \[3\] Jesse Mu, et al. "Improving Intrinsic Exploration with Language Abstractions." In NeurIPS 2022.

---

> > ### Comment · Reviewer_qGp8 · 2023-11-20
> > **Quick MiniHacks follow-up questions**
> >
> > Thank you so much for your responses. I have read through your comments on this thread as well as the discussion going on in other threads about MiniHacks. These added experiments address many concerns I had with BabyAI. I only have two remaining quick questions:
> > 1. How is sub-task completion detected in MiniHacks, and are there any pre-defined sub-tasks?
> > 2. How does the evaluator generate intrinsic rewards in MiniHacks environment?

---

> ### Author Response · Authors · 2023-11-21
> **Response to the follow-up questions**
>
> Thank you for your prompt response. We are happy to hear that we have addressed your main concerns. For your remaining questions, we will provide point-to-point responses:
> > How is sub-task completion detected in MiniHacks, and are there any pre-defined sub-tasks?
>
> **Response:** To be precise, there are no **man-made** pre-defined sub-tasks in MiniHacks. Actually, we employ a general and easy-to-extend approach to automate the process of the Planner acquiring and using knowledge from the environment, which is **using the system text messages in the environment as the source of knowledge**. Therefore, the completion of these sub-tasks is entirely detected within the game (automatically determined by the built-in trigger logic of messages designed by the game developers). Also, these text messages are embedded in the input observations for PAE and other baselines, while the superior experimental performance of PAE demonstrates that PAE is the best to effectively utilize these textual knowledge, achieving the fastest convergence speed and the best performance.
>
> In fact, the detection of subtasks is also automatically handled by BabyAI's built-in game logic. The only difference is that, **BabyAI's subtasks appear more structured and template-like** in their message representation and are closer to what humans would imagen as "instructions".
>
> > How does the evaluator generate intrinsic rewards in MiniHacks environment?
>
> **Response:** For the Evaluator, we used the same hyperparameters and the same evaluating logic (Section 4.3 of in our main paper). This, to some extent, may also justify the scalability of our method. Besides, we must acknowledge that our experiments on Minihack were conducted within very limited time. Theoretically, more detailed hyperparameter tuning could lead to even better performance.
>
> Thank you very much for your contribution in clarifying our ideas. Please be free to let us know if you have any further questions. Besides, if you find that these concerns have been resolved, we would be grateful if you would consider reflecting this in your rating of our paper : ) .

---

> > ### Comment · Reviewer_qGp8 · 2023-11-21
> >
> > Thank you for the clarifications, I've enjoyed the paper as well as the insightful discussions. It must have been a stretch running all these experiments in a short time. I have revised my rating to an 8!

---

> > > ### Author Response · Authors · 2023-11-21
> > > **Love and Best Wishes from the authors**
> > >
> > > We deeply appreciate the effort and time you've invested in the discussions, and we are delighted to receive your approval! Thank you for recognizing our efforts, and we also believe that the hard work we have done is worthwhile.
> > >
> > > The above inspiring discussion has greatly improved the quality of our paper. Thank you very much. We also enjoy the inspiring and insightful discussions with you!
> > >
> > > -- Love and Best Wishes from the authors ^_^

---

### Author Response · Authors · 2023-11-18
**Response to all reviewers**

We thank all the reviwers for the insightful, constructive, and helpful reviews. We are pleased that reviewers found our paper to be:

(1) well-written: "**well structured**" (qGp8), "**clear objectives**" (2pSJ), "**writing is clear**" (umQ7);
(2) contributing to the community: "**provides meaningful insights**" (2pSJ), "**very interesting and novel**" (rZvG), "**a valuable contribution to the RL community**" (2pSJ);
(3) technically sound: "**have consistent improvement**" (qGp8), "**provide strong evidence**" (2pSJ), "**strong performance**" (umQ7).

Overall, the reviewers' concerns primarily focused on scalability, summarized in terms of:

● Specific environment.

● Templated knowledge.

To address the above concerns, we conducted supplementary experiments on **five tasks in the more challenging MiniHack environment** during the limited rebuttal period. These experiments demonstrate that:

● The PAE method **adapts to more challenging environments** with complex state-action spaces. ([Screenshot of Minihack](https://ibb.co/sK1n2Fb)), ([Screenshot of Training-Curves](https://imgbb.com/QJKqdxP)), ([Screenshot of Addtional-Results](https://ibb.co/Wnsf0Qw))

● The PAE method **does not depend on specific knowledge templates**. ([Screenshot of Minihack Language](https://ibb.co/mDzz6Rm))

Furthermore, we responded to each reviewer's questions or concerns with point-to-point responses and clarifications or additional experimental support.

We have carefully read each comment and made revisions and improvements to our paper (see revision version). We are very grateful for the sincere feedback and contributions of the reviewers to our paper. Currently, the quality of our papers has markedly improved. We appreciate the opportunity to address the concerns raised and hope that our revisions have satisfactorily addressed these points. We remain open to any further feedback and are committed to making additional improvements if needed.

We sincerely hope that our revisions have adequately addressed your concerns. If you find that these concerns have been resolved, we would be grateful if you would consider reflecting this in your rating of our paper.

---

### Meta-Review · Area_Chair_LmVx · 2023-12-05

**Metareview:**

The paper introduces a a structured approach to exploration and information gathering in a new environment with a Planner-Actir-Evaluator framework. The framework, experiments and clarity were strengths pointed out by reviewers, weaknesses include the small and bespoke env and action space (that doesn't use the flexibility of an LM) -- starting off in a small env is ok for a new method. Over the course of a productive discussion period, all reviewers agree that this paper should be accepted.

**Justification For Why Not Higher Score:**

the small and bespoke env and action space (that doesn't use the flexibility of an LM)

**Justification For Why Not Lower Score:**

framework, experiments and clarity, relevance

---

### Decision · Program_Chairs · 2024-01-16

Accept (poster)